# Rewiring Neurons in Non-Stationary Environments

**Zhicheng Sun, Yadong Mu**[*]
Peking University, Beijing, China
{sunzc,myd}@pku.edu.cn

## Abstract

The human brain rewires itself for neuroplasticity in the presence of new tasks. We are inspired to harness this key process in continual reinforcement learning, prioritizing adaptation to non-stationary environments. In distinction to existing rewiring approaches that rely on pruning or dynamic routing, which may limit network capacity and plasticity, this work presents a novel rewiring scheme by permuting hidden neurons. Specifically, the neuron permutation is parameterized to be end-to-end learnable and can rearrange all available synapses to explore a large span of weight space, thereby promoting adaptivity. In addition, we introduce two main designs to steer the rewiring process in continual reinforcement learning: first, a multi-mode rewiring strategy is proposed which diversifies the policy and encourages exploration when encountering new environments. Secondly, to ensure stability on history tasks, the network is devised to cache each learned wiring while subtly updating its weights, allowing for retrospective recovery of any previous state appropriate for the task. Meanwhile, an alignment mechanism is curated to achieve better plasticity-stability tradeoff by jointly optimizing cached wirings and weights. Our proposed method is comprehensively evaluated on 18 continual reinforcement learning scenarios ranging from locomotion to manipulation, demonstrating its advantages over state-of-the-art competitors in performance-efficiency tradeoffs. Code is available at `https://github.com/feifeiobama/RewireNeuron`.

## 1 Introduction

Despite remarkable advances in artificial neural networks, they have yet to emulate the adaptivity of the human brain in realistic, non-stationary environments. A plausible explanation for this disparity from neuroscience is that the brain gains additional structural plasticity by constantly rewiring itself throughout one's lifetime [9, 35, 67]. In fact, with extensive rewiring, the brain could even confront more adverse scenarios such as stroke [11, 68]. By contrast, the wiring of artificial neural networks is usually optimized for a single domain [79, 72, 70] and remains fixed thereafter, thus exhibiting limited adaptivity in the real world. This comparison highlights the potential of rewiring neurons in enabling neural networks to continually engage and learn in dynamically evolving environments, which is referred to as the problem of continual reinforcement learning [53, 64].

A few preliminary studies have exploited this rewiring process in deep learning with other research scopes. They rewired neural networks through pruning [37, 28, 5] or dynamic routing [43, 55, 74] to allow more efficient or input-dependent inference at test time. Following this line of work, a mainstay of continual learning methods [41, 32, 23] learned task-specific masks on neuron wiring to alleviate catastrophic interference [42] between different tasks. Unfortunately, existing rewiring approaches that rely on selecting a subnetwork are typically prone to capacity shrinkage over time and therefore may not be competent for continual adaptation to new reinforcement environments.

---

[*]Corresponding author.

37th Conference on Neural Information Processing Systems (NeurIPS 2023).

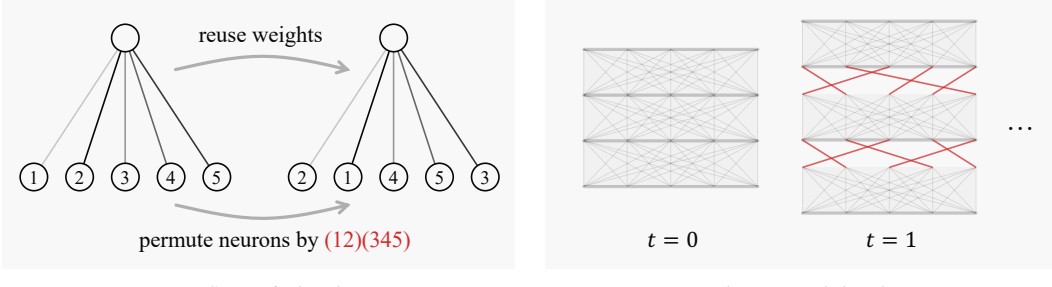

|  |  |
|:---:|:---:|
| (a) Synaptic level | (b) Network level |

Figure 1: Illustration of rewiring neurons via permutation. (a) In micro, permuting neurons allows reusing existing synaptic connections to create functionally new computation units. (b) In macro, layerwise permutation of hidden neurons provides extra structural plasticity in continual learning. Compared to alternative approaches, this method enjoys the higher capacity of the full network.

To harness the adaptivity of the rewiring process as in the human brain, we propose a novel rewiring scheme by neuron permutation, which fully reuses network weights to achieve structural plasticity. A motivating example is presented in Figure 1a, where instead of modifying synaptic strengths, one can directly permute neurons to populate certain desired functionalities. In more general cases, neuron permutation may serve as a complementary mechanism to weight updates to facilitate the learning process, as it encodes numerous variations by rewiring existing synapses. For instance, a tiny neural network shown in Fig. 1b encapsulates over 10,000 wirings to be induced by permutation, with the number growing exponentially w.r.t. network depth and even more dramatically w.r.t. width. And thanks to the recent differentiable sorting algorithms [22, 50], it can be efficiently parameterized for end-to-end learning and thus integrated for adaptation to new scenarios.

In the context of continual reinforcement learning, we introduce two rewiring designs to emphasize exploration and stability-plasticity, respectively, which are the two essential aspects of the problem. To enhance exploration in novel environments, a multi-mode rewiring strategy is devised. In detail, it promotes policy diversity by randomly selecting a rewiring mode at each step, while effectively sharing knowledge across different modes via a distillation loss. To enforce stability on past tasks, we let the network cache each learned wiring and regularize its weight changes, allowing previous states to be recovered at low memory cost. Meanwhile, an alignment mechanism is proposed that jointly refines cached wirings and the latest weights, complementing plasticity.

In summary, our contributions are as follows. (1) We propose a novel rewiring scheme by permuting hidden neurons, allowing for additional structural plasticity under non-stationarity. (2) To encourage exploration in new environments, a multi-mode rewiring strategy is introduced that increases policy diversity. (3) To ensure stability on history tasks, the network caches each learned wiring and aligns it with the newest weights. (4) Experiments on 18 continual reinforcement learning scenarios from Brax and Continual World clearly demonstrate the strengths of our method.

## 2 Related work

**Permuting neurons.** The effectiveness of neuron permutation has been demonstrated in aligning differently trained neural networks [38, 59]. Subsequent literature [6, 62, 16, 2] exploited permutation symmetries of hidden neurons [29] for finding low-loss connectors [13, 19] between loss minima. Further applications of permutation symmetry include designing equivariant neural architectures [46, 77] and facilitating continual learning [47]. Our work is orthogonal to the aforementioned studies in that we use permutation to produce functionally diverse networks, rather than simply aligning them.

The key technique behind permutation learning is differentiable sorting. A number of works [1, 44, 10, 15] approximated permutation with doubly stochastic matrices and ran the Sinkhorn algorithm [60]. Petersen *et al*. [48, 49] cast differentiable sorting as a series of continuous swap operations. However, both methods involve multiple forward iterations, rendering them inefficient for longer permutations. Therefore, a more lightweight alternative via unimodal row stochastic relaxation [22, 50] is adopted in our experiments. Among various efforts to embed this technique into the network structure [63, 56], we are the first to advocate it for promoting plasticity in continual learning.

**Continual learning.** To address the major issues in continual learning [53] (or lifelong learning [64]), such as catastrophic forgetting [42] and the stability-plasticity dilemma [7], three main approaches have been developed: memory replay [51, 58], weight regularization [33, 39] and parameter isolation [54, 41, 32, 23]. Within the last category, Mallya and Lazebnik [41] pioneered the use of pruning [37, 28] to freeze important parameters for stability, but at the cost of capacity shrinkage. Ke *et al*. [32] resorted to dynamic routing [43, 55] for sharing knowledge between different tasks. Gurbuz and Dovrolis [23] iteratively pruned and grew neuron connections in a sparse neural network to maintain fixed capacity. In comparison, our work leverages all model parameters with no additional pruning step, and can therefore be easily adapted to different scenarios.

**Continual reinforcement learning.** Since the earliest proposals [53, 64], continual learning has been intertwined with reinforcement learning [61] for control and robotics. In this intersecting area, a main stream of studies [36, 8, 71] made specific assumptions on the Markov decision process to theoretically maximize reward amidst non-stationarity. Several notable works [3, 45, 52] employed meta-learning [57] for faster adaptation to new environments. There are also counterparts [54, 31, 20] that focused on continual learning of policy networks at the parameter level. Along this direction, Rusu *et al*. [54] proposed to dynamically grow the policy network for incorporating new knowledge. Kaplanis *et al*. [31] distilled knowledge [30] from previously learned policies to ensure stability. Recently, Gaya *et al*. [20] suggested that learning a subspace of policies strikes a good balance between model size and performance. Following these efforts, we continue to push the performance-efficiency frontier with the proposed neuron rewiring scheme.

## 3 Method

In this section, we begin with a brief formulation of continual reinforcement learning (Section 3.1). Then, a novel rewiring approach is introduced in Section 3.2, along with its emphasis on two aspects: exploration (Section 3.3) and stability-plasticity (Section 3.4), both deemed crucial to the problem. Finally, we remark its connection to neuroscience observations in Section 3.5.

### 3.1 Problem formulation

Continual reinforcement learning concerns learning over non-stationary environments, which are characterized by a sequence of $T$ tasks. The goal is to obtain a well-performed global policy $\pi(\mathbf{a}|\mathbf{s}, t)$ that takes a state $\mathbf{s}$ and a task identifier $t \in [1 .. T]$ as input, and outputs a distribution over actions $\mathbf{a}$. To achieve this, a policy network is employed. Following the current literature [69, 20], we adopt a multi-layer perceptron architecture comprising a series of alternating affine layers $\boldsymbol{W} = \{\boldsymbol{W}_l\}_{l=1}^{L}$ [1] and element-wise activation functions $\sigma$. The policy inference is formulated as:

$$\boldsymbol{Y} = \boldsymbol{W}_L \circ \sigma \circ \boldsymbol{W}_{L-1} \circ \ldots \circ \sigma \circ \boldsymbol{W}_1 \boldsymbol{X}, \tag{1}$$

where $\boldsymbol{X}$ and $\boldsymbol{Y}$ denote the input and the predicted policy, respectively. This problem poses significant challenges on the policy network, as it must quickly adapt to environmental changes through efficient exploration, while not catastrophically forgetting the previously learned policy.

We address the problem from a structural perspective, focusing on the design of the policy network. Unlike many existing approaches of this type that grow separate policy networks for each task at linearly increasing memory cost, our solution involves only two networks: one for the current task and one for the previous task. We will refer to the current network by default, and when necessary, use the superscripts $t$ and $t - 1$ to distinguish between these two networks.

### 3.2 Rewiring via permutation

Inspired by the brain's remarkable adaptivity by rewiring itself [9, 35, 67], we seek to incorporate a similar process into our policy network. Ideally, this would provide additional plasticity to standard weight learning in non-stationary environments. However, the implementation is not trivial due to major architectural differences between the brain and our network. Most notably, the brain exhibits sparse connectivity, whereas our network consists of fully connected layers.

---

[1] For simplicity, we will omit biases from our presentation, while it is fairly easy to extend the following discussion to include biases.

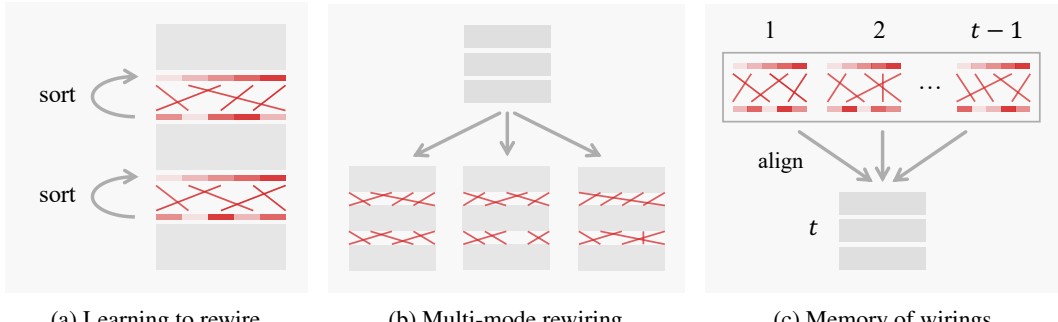

| (a) Learning to rewire | (b) Multi-mode rewiring | (c) Memory of wirings |

Figure 2: Illustration of our proposed rewiring framework. (a) The rewiring process, formulated as neuron permutation, is parameterized by score vectors indicating the neuron order within each layer, which can be learned via a differentiable sorting algorithm. (b) In the presence of a new task, our network simultaneously learns $K$ different wirings to diversify the policy and promote exploration. (c) Each learned wiring is cached and then continually refined to align with the newest weights.

To this end, we propose a novel rewiring scheme that exploits the layered structure of the network. While each layer has intricate connections and rewiring within it may seem complex, we take an alternative approach by rewiring between layers through neuron permutation, as shown in Fig. 1b. This allows reusing existing synapses to redirect the information flow within the network, thereby inducing new wirings with varying functionalities. Formally, it is described as inserting a series of permutation matrices $\boldsymbol{P} = \{\boldsymbol{P}_l\}_{l=1}^{L-1}$ into the network architecture:

$$\boldsymbol{Y} = \boldsymbol{W}_L \circ \sigma \circ \boldsymbol{P}_{L-1}\boldsymbol{W}_{L-1} \circ \ldots \circ \sigma \circ \boldsymbol{P}_1\boldsymbol{W}_1\boldsymbol{X}. \tag{2}$$

Note that we permute only the hidden neurons at layer $l \in [1 .. L-1]$, since the input and output formats are assumed to be fixed. This formulation also differs from permutation symmetry works [62, 16, 2], as our focus is on populating different wirings rather than equivalent ones.

Then, a differentiable sorting algorithm is employed to learn the permutations, as illustrated in Fig. 2a. We follow Prillo and Eisenschlos [50] to parameterize each permutation $\boldsymbol{P}_l$ with a score vector $\boldsymbol{v}_l$. In detail, this score vector $\boldsymbol{v}_l$ uniquely corresponds to an order $\boldsymbol{z}_l$, which can then be mapped bijectively to a permutation matrix. Thus, the permutation can be computed in a forward pass:

$$\boldsymbol{P}_l = \boldsymbol{I}[\boldsymbol{z}_l, :], \quad \boldsymbol{z}_l = \mathrm{argsort}(\boldsymbol{v}_l), \tag{3}$$

where $\boldsymbol{I}$ is an identity matrix. Since both the indexing and argsort operators are non-differentiable, the following smoothed proxy is used instead during backpropagation:

$$\hat{\boldsymbol{P}}_l = \mathrm{softmax}\left(\frac{-d(\mathrm{sort}(\boldsymbol{v}_l)\mathbf{1}^\top, \mathbf{1}\boldsymbol{v}_l^\top)}{\tau}\right), \tag{4}$$

where $d(\cdot, \cdot)$ uses the $L_1$ distance and $\mathbf{1}$ is an all-ones vector. This way we can backpropagate loss errors through permutation and learn the score vector $\boldsymbol{v}_l$ in an end-to-end manner. For clarity, the following text will represent the induced wiring as $\boldsymbol{P}_l$, omitting parameterization details involving $\boldsymbol{v}_l$.

In addition, we would like to highlight two aspects of the proposed rewiring scheme:

**Parameter efficiency.** Neuron permutation is highly parameter-efficient. In a network of width $n$, each layer of permutation learning requires only $O(n)$ parameters. And after training, they can be compressed into a discrete order with even smaller memory footprint. This distinguishes our method from incorporating additional linear layers, which introduces a quadratic parameter overhead that is undesirable in a resource-constrained continual learning setting.

**Adaptivity.** Neuron permutation exploits a variety of structural variations to facilitate adaptation. There are over $10^{1520}$ potential wirings in our adopted policy network, and the number continues to grow factorially w.r.t. network width and exponentially w.r.t. depth. Moreover, each wiring maps the current network weight to a distinct weight point, thus covering a large span of weight space. Together with the parameter efficiency, it significantly enhances the adaptivity of our network in non-stationary environments, which will be validated through experiments.

## 3.3 Rewiring for exploration

In applying the proposed rewiring scheme to continual reinforcement learning, there are two key challenges that need to be addressed. The first challenge is to efficiently explore the policy space when encountering a novel environment, in order to avoid being trapped in suboptimal solutions. This notion of exploration has been emphasized in many related areas, such as maximum entropy reinforcement learning [78, 24, 25], unsupervised skill discovery [17], and few-shot adaptation [34, 21].

To encourage exploration in new environments, we propose a multi-mode rewiring strategy that maintains a set of different wirings to induce diverse policies, as shown in Fig. 2b. At each step, the agent randomly samples a mode to select from the following candidate wirings:

$$\boldsymbol{P}_l \in \{\boldsymbol{P}_{l,1}, \boldsymbol{P}_{l,2}, \dots, \boldsymbol{P}_{l,K}\}. \tag{5}$$

where $K$ is the total number of modes, and the mode is shared across all layers. Also, to ensure an equivalent learning rate, we multiply the learning rate of each wiring by $K$.

Figure 3 illustrates the policies induced by multi-mode rewiring. Interestingly, these policies naturally exhibit a diverge-and-converge pattern without any constraints on the wirings. That is, they diverge rapidly at first, but then gradually converge to a high similarity. This implies that multi-mode rewiring enables the agent to explore various policies before finally settling on some effective ones. Thus, we can safely select one of the wirings for subsequent evaluation and training.

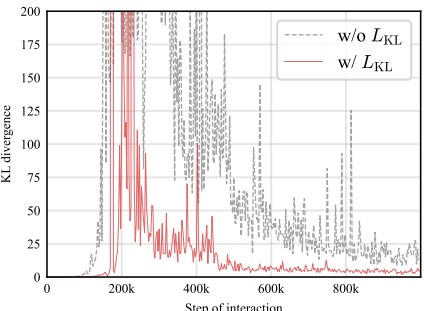

We further distill knowledge [30] across different wirings for knowledge sharing, which resembles the idea of mutual learning [76, 27] but in a parameter-efficient manner. Kullback-Leibler (KL) divergence is used as the loss:

Figure 3: Evolution of Kullback-Leibler divergence between different policies induced by multi-mode rewiring at $K = 3$.

$$L_{\mathrm{KL}}(\boldsymbol{W}, \boldsymbol{P}) = \mathbb{E}_{k' \neq k} \left[ D_{\mathrm{KL}} \left( \pi_{k'}(\cdot|\mathbf{s}) \| \pi_k(\cdot|\mathbf{s}) \right) \right], \tag{6}$$

where $\pi_k$ is the policy induced by the $k$-th wiring, and we only backpropagate through one wiring to save computational cost. As shown in Fig. 3, it significantly speeds up the convergence within modes.

## 3.4 Rewiring for stability-plasticity

The second major challenge in continual reinforcement learning is the stability-plasticity dilemma [7], where an agent could either suffer severe performance degradation on history tasks due to catastrophic interference [42], or fail to adapt to new environments if an overly rigid constraint is imposed to mitigate forgetting. This requires careful maintenance of old policies when acquiring new ones.

**Caching each wiring.** Our base solution is to cache each learned wiring and regularize the weights, as shown in Fig. 2c. Intuitively, this would allow recovery of both previous weights and wirings, thereby preserving the learned policies. On the wiring side, thanks to the parameter efficiency of permutation, it yields a very small storage overhead (less than 0.5% per wiring in our network) compared to caching the entire network. On the weight side, a typical choice would be $L_2$ regularization, which has proven to mitigate forgetting well in many continual reinforcement learning scenarios [69, 20]:

$$L_{\mathrm{reg}}(\boldsymbol{W}^t) = \sum_{l=1}^{L} \|\boldsymbol{W}_l^t - \boldsymbol{W}_l^{t-1}\|^2. \tag{7}$$

However, such a scheme with weight regularization is usually prone to the stability-plasticity dilemma mentioned above, since the plasticity of the weights is largely reduced by the weight regularizer.

**Aligning wirings with weights.** To address the stability-plasticity dilemma in weight regularization, we reformulate the problem into the misalignment between the latest weights and cached wirings. For example, the forgetting since the last training stage can be reproduced by substituting the newest weights $\boldsymbol{W}_l^t$ and the last cached wiring $\boldsymbol{P}_l^{t-1}$ into the inference procedure in Eq. (2):

$$\boldsymbol{Y} = \dots \circ \sigma \circ \boldsymbol{P}_l^{t-1} \boldsymbol{W}_l^t \circ \dots \boldsymbol{X}. \tag{8}$$

Instead of framing it as a shift in the weight space (as in EWC [33], SI [75], *etc.*), we view it as the misalignment between updated weights and fixed wiring. This provides a new perspective on the loss of plasticity: existing schemes always align the weights to the wiring, while the wiring remains fixed, resulting in an overly rigid constraint. So we ask, why not align the wiring to the weights?

Inspired by this, we propose a new alignment mechanism that jointly refines the wiring and the weights to align with each other. Specifically, the optimization of each cached wiring is achieved by inserting two wiring adapters $\boldsymbol{P}_l'$ and $\boldsymbol{P}_l''$ before and after it. In this way, the wiring in Eq. (8) is adapted to:

$$\boldsymbol{Y} = \ldots \circ \sigma \circ \underbrace{\boldsymbol{P}_l' \boldsymbol{P}_l^{t-1} \boldsymbol{P}_l''^{\top}}_{\text{adapters on } \boldsymbol{P}_l^{t-1}} \boldsymbol{W}_l^t \circ \ldots \boldsymbol{X}. \tag{9}$$

Note that the element-wise activation functions can be interchanged with permutation layers, yielding an equivalent form in which the adapters could alternatively be interpreted as adapters on the weights:

$$\boldsymbol{Y} = \ldots \circ \sigma \circ \boldsymbol{P}_l^{t-1} \underbrace{\boldsymbol{P}_l''^{\top} \boldsymbol{W}_l^t \boldsymbol{P}_{l-1}'}_{\text{adapters on } \boldsymbol{W}_l^t} \circ \ldots \boldsymbol{X}. \tag{10}$$

From here, we can switch back from the misalignment interpretation to the weight shift interpretation. In order to mitigate forgetting, the adapted weights should be close to the previous weights. which yields the following regularizer on both the wiring and the weights:

$$L_{\text{SP}}(\boldsymbol{W}^t, \boldsymbol{P}', \boldsymbol{P}'') = \sum_{l=1}^{L} \|\boldsymbol{W}_l^t - \boldsymbol{P}_l'' \boldsymbol{W}_l^{t-1} \boldsymbol{P}_{l-1}'^{\top}\|^2. \tag{11}$$

And after the wiring adapters are learned, they are used to refine all cached wirings. For instance, the last cached wiring is updated as:

$$\boldsymbol{P}_l^{t-1} = \boldsymbol{P}_l' \boldsymbol{P}_l^{t-1} \boldsymbol{P}_l''^{\top}. \tag{12}$$

Replacing the previous weight regularizer with this new alignment mechanism improves stability-plasticity in two aspects: (1) The weight constraint is relaxed, since the weight is not necessarily constrained near the previous one. (2) With aligned wirings to counteract weight changes, a smaller coefficient on the regularizer will suffice to mitigate forgetting.

Finally, the proposed rewiring designs are incorporated into the existing reinforcement learning framework (Soft Actor-Critic [25] in our case) using a weighted sum of the standard training objective (maximum reward plus maximum entropy) and the two newly introduced losses $L_{\text{KL}}$ and $L_{\text{SP}}$ with coefficients $\alpha$ and $\beta$. The rest of the training specifications remain unchanged.

### 3.5 Connection to neuroscience

Since our inspiration comes from the rewiring process in the brain, it is natural to review and compare our designs with relevant neuroscience observations. Here, we would like to remark some noteworthy similarities and distinctions between our rewiring approach and the actual process.

**Interdependence of weights and wiring.** It is elucidated that the weight and wiring changes in the brain are not mutually exclusive, but could take place simultaneously to facilitate plasticity [9]. In our approach, a similar line of thinking is followed by updating both weights and wiring during adaptation to new environments. Moreover, we introduce an alignment mechanism (Eqs. (11) and (12)) that effectively coordinates these two updates to promote stability-plasticity.

**Layerwise locality.** *In vivo* imaging shows that long-range axonal connections are remarkably stable, while local dendritic synapses are more plastic [66]. This suggests that rewiring neighboring neurons may be the preferred approach. Given the layered structure of artificial neural networks, we adhere to rewiring neurons within the same layer. It should also be noted that there are other levels of locality to consider [40], but we find that incorporating layerwise locality already suffices.

**Absence of sparsity.** The main difference between our method and brain rewiring is that we opt for a dense and intensive rewiring scheme, whereas the brain rewires neurons sparsely [9]. This is due to the fact that the human brain has a vastly greater number of neurons (estimated at 86 billion [4]) compared to the neural network we used (with hundreds of neurons). Hence, to adequately address the problem, we need to leverage all available connections to maximize capacity.

Table 1: Average performance (↑) and model size (↓) on Brax scenarios across HalfCheetah, Ant and Humanoid. The results on HalfCheetah and Ant are averaged over 4 different scenarios, and you may find the full results in Appendix B.1. The baseline performances are summarized by Gaya *et al*. [20]. Our method achieves near state-of-the-art performance with less than 60% of the parameters.

| Method | HalfCheetah | | Ant | | Humanoid | |
|---|---|---|---|---|---|---|
| | Performance | Model size | Performance | Model size | Performance | Model size |
| FT-1 | $0.62 \pm 0.29$ | **1.0** | $0.52 \pm 0.26$ | **1.0** | $0.71 \pm 0.07$ | **1.0** |
| FT-L2 | $0.38 \pm 0.15$ | 2.0 | $0.78 \pm 0.20$ | 2.0 | $0.68 \pm 0.28$ | 2.0 |
| PackNet [41] | $0.85 \pm 0.14$ | 2.0 | $1.08 \pm 0.21$ | 2.0 | $0.96 \pm 0.21$ | 2.0 |
| EWC [33] | $0.43 \pm 0.24$ | 3.0 | $0.55 \pm 0.24$ | 3.0 | $0.94 \pm 0.01$ | 3.0 |
| PNN [54] | $1.03 \pm 0.14$ | 8.0 | $0.98 \pm 0.31$ | 8.0 | $0.98 \pm 0.26$ | 4.0 |
| SAC-N | $1.00 \pm 0.15$ | 8.0 | $1.00 \pm 0.38$ | 8.0 | $1.00 \pm 0.29$ | 4.0 |
| FT-N | $1.16 \pm 0.20$ | 8.0 | $0.97 \pm 0.20$ | 8.0 | $0.65 \pm 0.46$ | 4.0 |
| CSP [20] | **1.27 ± 0.27** | 5.4 | $1.11 \pm 0.17$ | 3.9 | $1.76 \pm 0.19$ | 3.4 |
| Ours | $1.17 \pm 0.15$ | 2.1 | **1.22 ± 0.11** | 2.1 | **1.78 ± 0.22** | 2.0 |

## 4 Experiments

### 4.1 Experimental settings

**Environments.** We use 18 continual reinforcement learning scenarios from Brax and Continual World: (1) Brax [18, 20] contains 9 locomotion scenarios over 3 domains: HalfCheetah, Ant and Humanoid. HalfCheetah and Ant each include 4 scenarios focused on forgetting, transfer, robustness, or compositionality. Each scenario has 8 tasks with different dynamics and up to 1 million interactions per task. For Humanoid, there is one scenario with 4 sequential tasks and a budget of 2M interactions per task. (2) Continual World [69] is a manipulation benchmark built on Meta-World [73] and MuJoCo [65], featuring 8 scenarios with 3 tasks (CW3) and one scenario with 10 tasks (CW10), both with a varying reward function and a budget of 1M interactions per task. More details are provided in Appendix A.1.

**Baselines.** Our method is compared to commonly used baselines including PNN [54], EWC [33], PackNet [41] and CSP [20], as well as simpler strategies such as finetuning a single model (FT-1), finetuning with $L_2$ regularization (FT-L2), training models from scratch for each task (SAC-N) and finetuning with checkpoints for each task (FT-N). Note that replay-based approaches are not included, since our formulation assumes no access to earlier replay buffers. The configurations for each method are described in Appendix A.2. We report the baseline results from [69, 20].

**Metrics.** Average performance and model size are selected as the evaluation criteria, following [20]. The first metric is defined on Brax as the cumulative reward normalized by the base result of SAC-N and on Continual World as the success rate, both averaged over all tasks seen at the end of training. For model size, we use the maximum number of parameters divided by that of a single policy (FT-1). Further indicators including forgetting and forward transfer are introduced in Appendix A.3. Unless otherwise noted, the new performances on Brax are averaged over 10 different seeds.

**Implementation details.** We build on the SaLinA library [12] and adopt Soft Actor-Critic (SAC) [25] with autotuned temperature [26] as the underlying algorithm. Both the actor and the critic are 4-layer perceptions with 256 hidden neurons per layer, while the actor also includes task-specific heads [69]. Their training configurations follow [20]. For our method, we choose the new hyperparameters $K$, $\alpha$, and $\beta$ via grid search for each scenario, and provide a sensitivity analysis in Section 4.3. The score vectors in Eq. (4) are initialized with an arithmetic sequence rescaled to $[0, 1]$, and the temperature is $\tau = 1$ by default. Furthermore, the computational cost is discussed in Appendix A.4.

### 4.2 Main results

The results on Brax are summarized in Table 1. Overall, our approach achieves strong results with a relatively small model size, demonstrating the effectiveness of rewiring neurons in continually adapting to new environments. Specifically, we observe from the results that: (1) Our method yields state-of-the-art performance on the Ant and Humanoid domains, with less than 60% of the parameters used by the previous best-performing method CSP. (2) On the HalfCheetah domain, where our

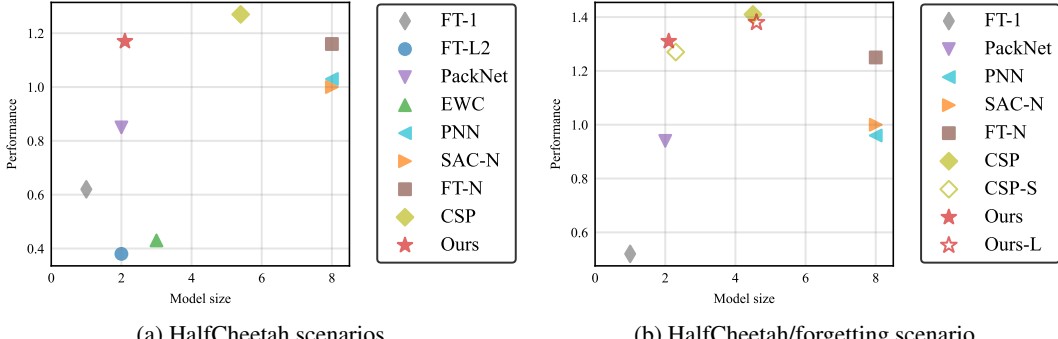

(a) HalfCheetah scenarios          (b) HalfCheetah/forgetting scenario

Figure 4: Performance-size tradeoffs for the relatively underperforming HalfCheetah scenarios. To compare at similar sizes, CSP-S reduces the network width to 175, while Ours-L expands it to 384. Our approach is on par with the leading method CSP and outperforms many of the larger baselines.

Table 2: Success rate(↑) on Continual World.

| Method | CW3 | CW10 |
|---|---|---|
| FT-1 | $0.29 \pm 0.10$ | $0.10 \pm 0.01$ |
| FT-L2 | $0.22 \pm 0.11$ | $0.48 \pm 0.05$ |
| PackNet [41] | $0.49 \pm 0.12$ | $0.83 \pm 0.02$ |
| EWC [33] | $0.32 \pm 0.11$ | $0.66 \pm 0.03$ |
| PNN [54] | $0.56 \pm 0.14$ | $0.63 \pm 0.07$ |
| SAC-N | $0.60 \pm 0.14$ | $0.73 \pm 0.12$ |
| FT-N | $0.65 \pm 0.11$ | $0.77 \pm 0.09$ |
| CSP [20] | $0.65 \pm 0.08$ | $0.81 \pm 0.06$ |
| Ours | $\mathbf{0.67 \pm 0.09}$ | $\mathbf{0.84 \pm 0.02}$ |

Table 3: Ablation studies of each rewiring design on the HalfCheetah/forgetting scenario. We start with the baseline method FT-L2 and add designs incrementally. * indicates rewiring without caching the wirings.

| Rewire | Multi-mode | Alignment | Performance |
|---|---|---|---|
|  |  |  | $0.67 \pm 0.32$ |
| * |  |  | $0.76 \pm 0.29$ |
| ✓ |  |  | $0.95 \pm 0.24$ |
| ✓ | ✓ |  | $1.14 \pm 0.22$ |
| ✓ |  | ✓ | $1.15 \pm 0.25$ |
| ✓ | ✓ | ✓ | $\mathbf{1.31 \pm 0.21}$ |

performance is comparatively lower, we manage to surpass multiple baseline methods (PNN, SAC-N and FT-N) that have a model size four times larger, and remain on par with the state-of-the-art method CSP using less than half the parameters. (3) Across all three domains, our method consistently outperforms existing baselines with a similar model size (PackNet and FT-L2) by large margins. (3) Across all three domains, our method consistently outperforms existing baselines.

In addition, Fig. 4 illustrates the performance-size tradeoff of different methods on HalfCheetah scenarios. While these are underperforming scenarios for our method, it still maintains a competitive performance-size tradeoff against alternative methods. In detail, our work outperforms models of comparable size (PackNet and FT-L2) by a large performance margin of more than 0.3, while also reducing the number of parameters by over 50% compared to the leading method CSP.

We further compare the performance on Continual World in Table 2. Moving to another continual reinforcement learning domain with manipulation tasks, our method consistently achieves state-of-the-art performance. The high success rates on both short 3-task scenarios and the longer 10-task scenario confirm the competence of our method in continually adapting to new tasks.

### 4.3 Ablation studies

The following analyses are performed on the HalfCheetah/forgetting scenario alone, due to the high time cost of conducting ablation experiments on all scenarios. We choose this scenario because it is relatively representative in terms of reflecting the overall performance, as indicated in Appendix B.1.

**Effectiveness of rewiring.** We first verify the effectiveness of rewiring neurons in improving network adaptivity when encountering new environments. This is achieved by comparing the performance evolution to that of the vanilla finetuning approach. As shown by the performance curve in Fig. 5a, our rewiring approach exhibits a substantial speedup in adaptation to an unseen scenario, reaching an average reward of 2000 four times faster than the baseline finetuning method. And afterward, this method achieves a significantly higher final reward over 3000, even without any additional design. These results clearly demonstrate the efficacy of the proposed rewiring approach.

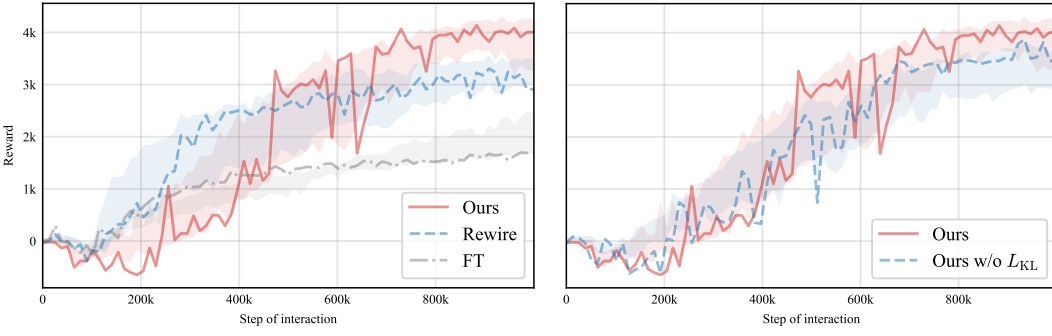

(a) Effectiveness of rewiring and multi-mode  (b) Effectiveness of the distillation loss $L_{\mathrm{KL}}$

Figure 5: Evolution of performance in the first stage of HalfCheetah/forgetting scenario. Ours denotes the full method, Rewire indicates rewiring without the multi-mode strategy, FT denotes finetuning. The curves depict the median, with shaded areas showing 95% bootstrap confidence interval around the mean. Our method yields the highest final performance, proving the effectiveness of each design.

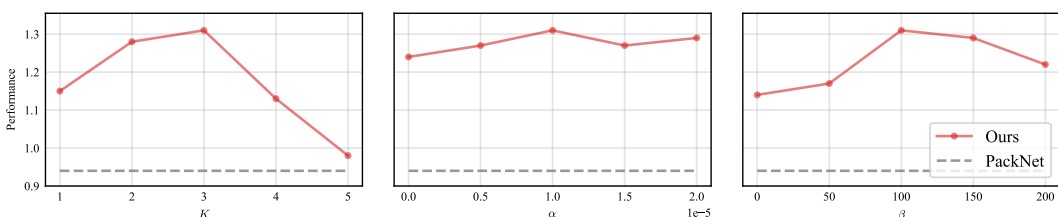

Figure 6: Trend of the mean performance (over 3 runs) w.r.t. the main hyperparameters: the number of modes $K$, the weight $\alpha$ for the distillation loss $L_{\mathrm{KL}}$, and the weight $\beta$ for the alignment loss $L_{\mathrm{SP}}$.

**Effectiveness of multi-mode.** The primary goal of multi-mode rewiring is to encourage exploration, which we validate with the performance curves in Fig. 5a. Although the full model exhibits slower initial learning around step 200k due to the divergence of multi-mode policies (as shown in Fig. 3), it ultimately outperforms upon convergence and achieves a final reward of nearly 4000. As a result, the inclusion of the multi-mode design increases the performance by over 0.16 in Table 3. We also compare the exploration efficacy of our multi-mode strategy to an existing exploration method called pink noise [14], with results presented in Appendix B.3 confirming the superiority of our approach. In addition, the ablation study in Fig. 5b shows that incorporating explicit convergence guidance improves both mean and median rewards, justifying the use of the distillation loss $L_{\mathrm{KL}}$.

**Effectiveness of memory.** This part of the model includes two designs: caching learned wirings, and aligning them with the weight. While both designs effectively address the problem of catastrophic forgetting, we regret that we cannot provide a detailed demonstration here, as the baseline FT-L2 method already achieves a high level of stability. On the other hand, these designs offer the advantage of plasticity, which is evident in the overall performance. According to the results presented in Table 3, the inclusion of the caching of each wiring is crucial for our method, as its removal would lead to a substantial performance degradation of about 0.19. Similarly, the alignment mechanism contributes significantly to the performance, improving it by more than 0.17. Further studies of the alignment mechanism by comparing the performance evolution are provided in Appendix B.3.

**Hyperparameter analysis.** Figure 6 depicts the impact of the newly introduced hyperparameters $K$, $\alpha$ and $\beta$ on the performance. Overall, regardless of the hyperparameter setting, our method consistently outperforms the baseline PackNet with a similar model size, demonstrating its non-sensitivity to hyperparameters. Specifically, for the number of modes $K$, the performance peaks at 3 and then declines. We suspect that the performance drop after that is related to the reduced number of optimization iterations assigned to each wiring. For the distillation weight $\alpha$, the performance increases stably within a reasonable range. For the alignment weight $\beta$, we mix it proportionally with the weight on the $L_2$ regularizer when the model is under-regularized, in order to have a fair comparison between the two regularizers. It can be seen that the performance increases significantly by incorporating our alignment mechanism, until the model becomes less plastic under a large weight.

## 5 Conclusion

In this work, we exploit the rewiring process of neural networks for continual reinforcement learning. Concretely, we propose the permutation of hidden neurons as a complement to the standard weight learning, and automate it with differentiable sorting algorithms. The learned permutation efficiently reuses all existing synapses to explore a multitude of weight variations, thus promoting plasticity. Furthermore, we introduce a multi-mode rewiring strategy and an alignment mechanism to address the challenges of exploration and stability-plasticity within the problem. Finally, the effectiveness of our method is validated on a variety of continual reinforcement learning benchmarks.

**Acknowledgement**:  This work is supported by National Key R&D Program of China (2022ZD0160305), and Beijing Natural Science Foundation (Z190001).

**Broader impact.** Our research contributes to the progress of biologically inspired neural networks and has potential uses in developing more efficient and adaptive AI systems for real-world environments. However, we recognize that safety-critical scenarios, such as autonomous driving, prioritize stability over other concerns, and a system as dynamic as ours may not be appropriate for these applications. It is essential to consider the adverse effects of implementing our method in such contexts.

**Limitations.** (1) The experimental setup does not cover replay-based continual learning strategies, nor more complex network architectures. As a result, the compatibility with them remains unclear. (2) Our method relies on task identifiers and cannot handle online task-free settings, which are closer to real scenarios but more difficult. (3) In future work, we also need to devise a more biologically plausible approach that modulates rewiring intensity throughout the learning process.

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
