# Rewiring Neurons in Non-Stationary Environments (Supplementary Material)

**Zhicheng Sun, Yadong Mu**[*]
Peking University, Beijing, China
{sunzc,myd}@pku.edu.cn

## A Experimental details

We conduct experiments using the open-source reinforcement learning library Salina [2], which is released under the MIT license. In the following, we provide more information about the environment details (Appendix A.1), method configurations (Appendix A.2), evaluation metrics (Appendix A.3), and computational costs (Appendix A.4).

### A.1 Environments

**Brax [4]** is a hardware-accelerated physics engine released under the Apache-2.0 license. To build a continual reinforcement learning benchmark on it, Gaya *et al*. [5] adapted three of its locomotion environments, including HalfCheetah (obs dim: 18, action dim: 6), Ant (obs dim: 27, action dim: 7), and Humanoid (obs dim: 376, action dim: 17), to derive varied environments. The resulting 26 tasks are summarized in Table 1.

For HalfCheetah, four scenarios are curated in [5] that focus on different aspects of continual learning, including a forgetting scenario where learning the next task tends to forget the previous one, a transfer scenario with negative forward transfer (see Appendix A.3 for definition) across tasks, a robustness scenario that alternates between a normal task and a distraction task, and a compositionality scenario where the final task is a combination of the previous variations. Specifically, each of them is composed of a 4-task sequence repeated twice:

1. Forgetting: hugefeet → moon → carrystuff → rainfall
2. Transfer: carrystuff_hugegravity → moon → defective_sensor → hugefeet_rainfall
3. Robustness: normal → inverted_action → normal → inverted_action
4. Compositionality: tinyfeet → moon → carrystuff_hugegravity → tinyfeet_moon

Similarly, Ant includes four different scenarios, each consisting of a 4-task sequence repeated twice:

1. Forgetting: normal → hugefeet → rainfall → moon
2. Transfer: nofeet_1_3 → nofeet_2_4 → nofeet_1_2 → nofeet_3_4
3. Robustness: normal → inverted_actions → normal → inverted_actions
4. Compositionality: nofeet_2_3_4 → nofeet_1_3_4 → nofeet_1_2 → nofeet_3_4

In addition, there is a humanoid scenario with higher observation and action dimensions. It consists of the following 4-task sequence: normal → moon → carrystuff → tinyfeet.

**Continual World [13]** is a continual reinforcement learning benchmark based on Meta-World [14], which is composed of 50 manipulation tasks originally curated for meta-reinforcement learning and is released under the MIT license. Underlying both benchmarks is MuJoCo [11], a general purpose physics engine released under the Apache-2.0 license.

---

[*]Corresponding author.

37th Conference on Neural Information Processing Systems (NeurIPS 2023).

Table 1: List of the 26 tasks used in Brax scenarios [4, 5] with their descriptions.

| | Task | Description |
|---|---|---|
| **HalfCheetah** | normal | - |
| | carrystuff | 4× mass and radius of the torso |
| | carrystuff_hugegravity | 4× mass and radius of the torso, 1.5× gravity |
| | defective_sensor | half observations are masked |
| | hugefeet | 1.5× mass and radius of the feet |
| | hugefeet_rainfall | 1.5× mass and radius of the feet, 0.4× friction |
| | inverted_actions | inverted action values |
| | moon | 0.15× gravity |
| | tinyfeet | 0.5× mass and radius of the feet |
| | tinyfeet_moon | 0.5× mass and radius of the feet, 0.15× gravity |
| | rainfall | 0.4× friction |
| **Ant** | normal | - |
| | hugefeet | 1.5× mass and radius of the feet |
| | nofeet_2_3_4 | only the 1st leg is enabled |
| | nofeet_1_3_4 | only the 2nd leg is enabled |
| | nofeet_1_3 | the 1st diagonal legs are disabled |
| | nofeet_2_4 | the 2nd diagonal legs are disabled |
| | nofeet_1_2 | forefeet are disabled |
| | nofeet_3_4 | hindfeet are disabled |
| | inverted_actions | inverted action values |
| | moon | 0.15× gravity |
| | rainfall | 0.4× friction |
| **Humanoid** | normal | - |
| | moon | 0.15× gravity |
| | carrystuff | 4× mass and radius of the torso and lower waist |
| | tinyfeet | 0.5× mass and radius of the feet |

There are three types of scenarios of different lengths introduced in the original paper [13], including 8 triplets (CW3), a longer 10-task sequence (CW10), and a 20-task sequence (CW20) from simply repeating CW10 twice. We follow [5] in using CW3 and CW10 scenarios for experiments. In detail, the CW3 scenarios are designed to have a large forward transfer from the first task to the third task, with the second task serving as a distraction. They include the following triplets:

1. push-v1 → window-close-v1 → hammer-v1
2. hammer-v1 → window-close-v1 → faucet-close-v1
3. window-close-v1 → handle-press-side-v1 → peg-unplug-side-v1
4. faucet-close-v1 → shelf-place-v1 → peg-unplug-side-v1
5. faucet-close-v1 → shelf-place-v1 → push-back-v1
6. stick-pull-v1 → peg-unplug-side-v1 → stick-pull-v1
7. stick-pull-v1 → push-back-v1 → push-wall-v1
8. push-wall-v1 → shelf-place-v1 → push-back-v1

Meanwhile, the CW10 scenario comprises the following 10-task sequence: hammer-v1 → push-wall-v1 → faucet-close-v1 → push-back-v1 → stick-pull-v1 → handle-press-side-v1 → push-v1 → shelf-place-v1 → window-close-v1 → peg-unplug-side-v1.

## A.2 Methods

This section describes the configuration of each method. We start with the architectural design shared by all methods, and then delve into specific hyperparameter settings.

**Architecture.** The actor and the twin critics all use a 4-layer perception with 256 neurons per layer, including a task-specific head for the actor. Leaky ReLU (with $\alpha = 0.2$) [8] is employed as the activation after each layer. Generally, our architecture is similar to the one used in [13], except that the layer normalization [1] after the first layer is removed, since it is not trivial to incorporate task-dependent normalized statistics into the proposed alignment mechanism.

Table 2: Hyperparameter values for each Brax scenario, selected via grid search following [5].

(a) HalfCheetah

| Method | Hyperparameter | Forgetting | Transfer | Robustness | Compositionality |
|---|---|---|---|---|---|
| FT-N | lr policy | 0.001 | 0.0003 | 0.001 | 0.0003 |
| | lr critic | 0.0003 | 0.0003 | 0.001 | 0.0003 |
| | reward scaling | 1. | 1. | 1. | 10. |
| | target output std | 0.1 | 0.05 | 0.1 | 0.1 |
| | policy update delay | 2 | 2 | 4 | 4 |
| | target update delay | 2 | 2 | 2 | 4 |
| FT-L2 | $L_2$ coefficient | $10^4$ | $10^0$ | $10^2$ | $10^2$ |
| EWC [6] | Fisher coefficient | $10^{-2}$ | $10^0$ | $10^{-2}$ | $10^0$ |
| CSP [5] | threshold | 0.1 | 0.1 | 0.1 | 0.1 |
| | repeat alpha | 100 | 20 | 20 | 100 |
| Ours | number of modes | 3 | 3 | 3 | 3 |
| | $L_{KL}$ coefficient | $10^{-5}$ | $10^{-5}$ | $10^{-5}$ | $10^{-5}$ |
| | $L_{SP}$ coefficient | $10^2$ | $10^{-1}$ | $10^0$ | $10^1$ |

(b) Ant

| Method | Hyperparameter | Forgetting | Transfer | Robustness | Compositionality |
|---|---|---|---|---|---|
| FT-N | lr policy | 0.001 | 0.001 | 0.001 | 0.0003 |
| | lr critic | 0.001 | 0.001 | 0.001 | 0.0003 |
| | reward scaling | 10. | 1. | 1. | 10. |
| | target output std | 0.05 | 0.05 | 0.1 | 0.1 |
| | policy update delay | 2 | 2 | 2 | 4 |
| | target update delay | 4 | 2 | 4 | 4 |
| FT-L2 | $L_2$ coefficient | $10^4$ | $10^0$ | $10^0$ | $10^2$ |
| EWC [6] | Fisher coefficient | $10^{-2}$ | $10^4$ | $10^2$ | $10^{-2}$ |
| CSP [5] | threshold | 0.1 | 0.1 | 0.1 | 0.1 |
| | repeat alpha | 100 | 100 | 100 | 20 |
| Ours | number of modes | 3 | 3 | 3 | 3 |
| | $L_{KL}$ coefficient | $10^{-5}$ | $10^{-5}$ | $10^{-5}$ | $10^{-5}$ |
| | $L_{SP}$ coefficient | $10^1$ | $10^{-1}$ | $10^{-1}$ | $10^0$ |

(c) Humanoid

| Method | Hyperparameter | Humanoid |
|---|---|---|
| FT-N | lr policy | 0.001 |
| | lr critic | 0.0003 |
| | reward scaling | 0.1 |
| | target output std | 0.1 |
| | policy update delay | 1 |
| | target update delay | 1 |
| FT-L2 | $L_2$ coefficient | $10^{-2}$ |
| EWC [6] | Fisher coefficient | $10^{-2}$ |
| CSP [5] | threshold | 0.1 |
| | repeat alpha | 100 |
| Ours | number of modes | 3 |
| | $L_{KL}$ coefficient | $10^{-6}$ |
| | $L_{SP}$ coefficient | $10^0$ |

Table 3: Computational efficiency on the HalfCheetah/forgetting scenario. Our method has a lower computational cost, despite the need for two forward passes (to compute the distillation loss $L_{\text{KL}}$).

|          | MACs (M) | Model size |
|----------|----------|------------|
| FT-1     | 0.14     | 1.0        |
| PNN [10] | 1.08     | 8.0        |
| CSP [5]  | 0.63     | 4.5        |
| Ours     | 0.48     | 2.1        |

**Baselines.** We follow the hyperparameter settings in [5], which are determined via grid search. Specifically, the common hyperparameters such as learning rate and reward scaling are set according to the performance of FT-N, while the remaining hyperparameter values are selected per method. Table 2 summarizes the hyperparameter setups. It can be seen that the regularization-based FT-L2 accommodates a large regularizer coefficient. In addition, the architecture-based methods PackNet [9] and PNN [10] are tuned to have a comparable model size to other baselines.

**Ours.** We grid search the newly introduced three hyperparameters for each scenario. Their settings on Brax are listed in Table 2. As can be seen, a relatively small $L_{\text{SP}}$ coefficient is used across most of the scenarios, and its effectiveness in mitigating forgetting will be validated in Tables 5 to 7. For Continual World, we tune our hyperparameters on the T6 scenario, where the baseline FT-L2 performs poorly, and use the results as a default for other scenarios.

## A.3 Metrics

In addition to the two metrics used in the main paper, including average performance and model size, we also adopt two additional metrics commonly used in continual learning [7]. The results evaluated using these metrics will be presented in Tables 5 to 7 and 9.

**Forward transfer** measures the knowledge transfer across tasks. Suppose there are a total of $T$ tasks. The test performance on task $j$ after the $i$-th training stage is denoted by $P_{i,j}$, and the performance by training only on task $i$ is denoted by $b_i$. Then, forward transfer is calculated as:

$$FT = \frac{1}{T} \sum_{t=1}^{T} P_{T,i} - b_i. \tag{1}$$

In general, a positive forward transfer indicates the ability to perform "zero-shot" learning by exploiting the previously learned knowledge [7], whereas a negative forward transfer indicates that model plasticity is severely reduced due to the learning algorithm used.

**Forgetting** measures the average performance degradation on each task after training on the entire task sequence. Using the previously defined notation, it is defined as:

$$F = \frac{1}{T} \sum_{t=1}^{T} P_{t,t} - P_{T,t}. \tag{2}$$

It is worth noting that this metric is not very useful in the context of continual reinforcement learning. As presented in [13, 5], the forgetting of baseline methods is usually very low and often close to 0. This is due to the use of a large regularization weight or multiple network checkpoints. In contrast, our method can achieve a similar level of stability with a much smaller regularization weight and less parameter overhead, thus promoting plasticity and efficiency.

## A.4 Computational costs

Our experiments are performed on Intel(R) Xeon(R) CPU cores (E5-2650 v4 @ 2.20GHz), and each run uses a single NVIDIA 2080Ti GPU. While the runtime varies depending on server conditions and task specifics, we estimate an average runtime of 30 hours for Brax scenarios, which is between the baseline methods FT-N and FT-L2 ($\approx$ 25 hours) and the previous leading method CSP ($\approx$ 35 hours). As for GPU memory consumption, our approach yields a slight increase (30%) over FT-L2 due to the extra permutation layers, but is still much more efficient than CSP ($>$ 100%). Further comparison using multiply-add operations (MACs) and model size is shown in Table 3. Overall, our rewiring approach is efficient in terms of both time and memory costs.

Table 4: Reference rewards for Brax scenarios [5]. They are obtained by the baseline method SAC-N, with hyperparameter values specified in Table 2.

| | Scenario | Task | Reward | Average reward |
|---|---|---|---|---|
| **Halfcheetah** | Forgetting | hugefeet | 2209 | 3125 |
| | | moon | 2982 | |
| | | carrystuff | 6309 | |
| | | rainfall | 1001 | |
| | Transfer | carrystuff_hugegravity | 7233 | 4921 |
| | | moon | 3599 | |
| | | defective_sensors | 5909 | |
| | | hugefeet_rainfall | 2942 | |
| | Robustness | normal | 4932 | 5383 |
| | | inverted_actions | 5833 | |
| | | normal | 4932 | |
| | | inverted_actions | 5833 | |
| | Compositionality | tinyfeet | 6311 | 4479 |
| | | moon | 3932 | |
| | | carrystuff_hugegravity | 6319 | |
| | | tinyfeet_moon | 1355 | |
| **Ant** | Forgetting | normal | 3752 | 2398 |
| | | hugefeet | 2841 | |
| | | rainfall | 1596 | |
| | | moon | 1401 | |
| | Transfer | nofeet_1_3 | 3021 | 2294 |
| | | nofeet_2_4 | 4119 | |
| | | nofeet_1_2 | 1014 | |
| | | nofeet_3_4 | 1021 | |
| | Robustness | normal | 3542 | 3871 |
| | | inverted_actions | 4199 | |
| | | normal | 3542 | |
| | | inverted_actions | 4199 | |
| | Compositionality | nofeet_2_3_4 | 770 | 475 |
| | | nofeet_1_3_4 | 641 | |
| | | nofeet_1_2 | 201 | |
| | | nofeet_3_4 | 288 | |
| | Humanoid | normal | 1958 | 1935 |
| | | moon | 1691 | |
| | | carrystuff | 2379 | |
| | | tinyfeet | 1711 | |

# B   Full results

## B.1   Brax

The full results on three Brax domains are summarized in Tables 5 to 8, after being normalized by the reference rewards in Table 4. They include a 95% confidence interval derived from 10 individual runs, as presented in Table 8. Our method consistently demonstrates competitive performance across many scenarios, even with a small model size. Compared to FT-L2 which mitigates forgetting well, our method achieves better plasticity through a smaller regularization weight. Our rewiring approach also significantly outperforms the pruning-based PackNet by fully exploiting the network parameters.

## B.2   Continual World

The detailed results on 8 triplet (CW3) scenarios are summarized in Table 9. Our approach achieves near state-of-the-art performance over all scenarios. Notably, we surpass the previous leading method CSP in 7 out of 8 scenarios, as well as consistently outperforming the regularization-based baselines FT-L2 and EWC and the pruning-based PackNet by large margins.

Table 5: Detailed results on 4 HalfCheetah scenarios. Baseline results are taken from [5]. New results are collected using 10 different seeds and presented with mean and standard deviation.

| | Method | Performance ↑ | Model size ↓ | Transfer ↑ | Forgetting ↓ |
|---|---|---|---|---|---|
| Forgetting | FT-1 | $0.52 \pm 0.08$ | $\mathbf{1.0 \pm 0.0}$ | $0.19 \pm 0.23$ | $0.67 \pm 0.19$ |
| | FT-L2 | $0.67 \pm 0.32$ | $2.0 \pm 0.0$ | $-0.34 \pm 0.30$ | $\mathbf{-0.01 \pm 0.00}$ |
| | PackNet [9] | $0.94 \pm 0.18$ | $2.0 \pm 0.0$ | $-0.07 \pm 0.17$ | $-0.00 \pm 0.00$ |
| | EWC [6] | $0.64 \pm 0.26$ | $3.0 \pm 0.0$ | $-0.27 \pm 0.31$ | $0.09 \pm 0.13$ |
| | PNN [10] | $0.96 \pm 0.15$ | $8.0 \pm 0.0$ | $-0.04 \pm 0.13$ | $0.00 \pm 0.00$ |
| | SAC-N | $1.00 \pm 0.10$ | $8.0 \pm 0.0$ | $-0.00 \pm 0.09$ | $-0.00 \pm 0.00$ |
| | FT-N | $1.25 \pm 0.24$ | $8.0 \pm 0.0$ | $0.25 \pm 0.23$ | $0.00 \pm 0.00$ |
| | CSP [5] | $\mathbf{1.41 \pm 0.07}$ | $4.5 \pm 2.0$ | $\mathbf{0.41 \pm 0.06}$ | $0.00 \pm 0.00$ |
| | Ours | $1.31 \pm 0.21$ | $2.1 \pm 0.0$ | $-0.08 \pm 0.21$ | $0.00 \pm 0.00$ |
| Transfer | FT-1 | $0.86 \pm 0.70$ | $\mathbf{1.0 \pm 0.0}$ | $0.52 \pm 0.62$ | $0.66 \pm 0.42$ |
| | FT-L2 | $-0.03 \pm 0.07$ | $2.0 \pm 0.0$ | $-1.00 \pm 0.03$ | $\mathbf{-0.03 \pm 0.04}$ |
| | PackNet [9] | $0.99 \pm 0.25$ | $2.0 \pm 0.0$ | $-0.01 \pm 0.24$ | $0.00 \pm 0.00$ |
| | EWC [6] | $-0.13 \pm 0.23$ | $3.0 \pm 0.0$ | $-1.13 \pm 0.21$ | $0.00 \pm 0.02$ |
| | PNN [10] | $1.05 \pm 0.14$ | $8.0 \pm 0.0$ | $0.04 \pm 0.13$ | $-0.00 \pm 0.00$ |
| | SAC-N | $1.00 \pm 0.15$ | $8.0 \pm 0.0$ | $-0.00 \pm 0.14$ | $-0.00 \pm 0.00$ |
| | FT-N | $1.39 \pm 0.34$ | $8.0 \pm 0.0$ | $0.39 \pm 0.33$ | $0.00 \pm 0.01$ |
| | CSP [5] | $\mathbf{1.95 \pm 0.83}$ | $4.9 \pm 1.1$ | $\mathbf{0.93 \pm 0.79}$ | $-0.01 \pm 0.03$ |
| | Ours | $1.42 \pm 0.19$ | $2.1 \pm 0.0$ | $0.34 \pm 0.19$ | $0.01 \pm 0.03$ |
| Robustness | FT-1 | $0.36 \pm 0.25$ | $\mathbf{1.0 \pm 0.0}$ | $-0.11 \pm 0.20$ | $0.53 \pm 0.25$ |
| | FT-L2 | $0.22 \pm 0.16$ | $2.0 \pm 0.0$ | $-0.79 \pm 0.15$ | $-0.00 \pm 0.00$ |
| | PackNet [9] | $0.65 \pm 0.11$ | $2.0 \pm 0.0$ | $-0.35 \pm 0.10$ | $0.00 \pm 0.00$ |
| | EWC [6] | $0.68 \pm 0.28$ | $3.0 \pm 0.0$ | $-0.31 \pm 0.23$ | $0.01 \pm 0.09$ |
| | PNN [10] | $\mathbf{1.14 \pm 0.10}$ | $8.0 \pm 0.0$ | $\mathbf{0.14 \pm 0.10}$ | $0.00 \pm 0.00$ |
| | SAC-N | $1.00 \pm 0.29$ | $8.0 \pm 0.0$ | $0.00 \pm 0.28$ | $0.00 \pm 0.00$ |
| | FT-N | $0.98 \pm 0.12$ | $8.0 \pm 0.0$ | $-0.02 \pm 0.11$ | $-0.00 \pm 0.00$ |
| | CSP [5] | $1.01 \pm 0.13$ | $7.4 \pm 0.5$ | $0.01 \pm 0.12$ | $\mathbf{-0.00 \pm 0.01}$ |
| | Ours | $1.07 \pm 0.12$ | $2.1 \pm 0.0$ | $-0.03 \pm 0.12$ | $0.02 \pm 0.01$ |
| Compositionality | FT-1 | $0.75 \pm 0.12$ | $\mathbf{1.0 \pm 0.0}$ | $-0.04 \pm 0.09$ | $0.22 \pm 0.11$ |
| | FT-L2 | $0.66 \pm 0.03$ | $2.0 \pm 0.0$ | $-0.35 \pm 0.03$ | $0.01 \pm 0.03$ |
| | PackNet [9] | $0.79 \pm 0.03$ | $2.0 \pm 0.0$ | $-0.21 \pm 0.03$ | $-0.00 \pm 0.00$ |
| | EWC [6] | $0.53 \pm 0.17$ | $3.0 \pm 0.0$ | $-0.34 \pm 0.09$ | $0.13 \pm 0.12$ |
| | PNN [10] | $0.97 \pm 0.16$ | $8.0 \pm 0.0$ | $-0.03 \pm 0.16$ | $0.00 \pm 0.00$ |
| | SAC-N | $1.00 \pm 0.05$ | $8.0 \pm 0.0$ | $-0.00 \pm 0.05$ | $-0.00 \pm 0.00$ |
| | FT-N | $\mathbf{1.01 \pm 0.09}$ | $8.0 \pm 0.0$ | $\mathbf{0.01 \pm 0.09}$ | $0.00 \pm 0.00$ |
| | CSP [5] | $0.69 \pm 0.09$ | $3.4 \pm 1.5$ | $-0.31 \pm 0.09$ | $0.00 \pm 0.00$ |
| | Ours | $0.88 \pm 0.09$ | $2.1 \pm 0.0$ | $-0.18 \pm 0.09$ | $\mathbf{-0.00 \pm 0.00}$ |
| Aggregate | FT-1 | $0.62 \pm 0.29$ | $\mathbf{1.0 \pm 0.0}$ | $0.14 \pm 0.29$ | $0.52 \pm 0.24$ |
| | FT-L2 | $0.38 \pm 0.15$ | $2.0 \pm 0.0$ | $-0.62 \pm 0.13$ | $\mathbf{-0.01 \pm 0.02}$ |
| | PackNet [9] | $0.85 \pm 0.14$ | $2.0 \pm 0.0$ | $-0.15 \pm 0.09$ | $0.00 \pm 0.00$ |
| | EWC [6] | $0.43 \pm 0.24$ | $3.0 \pm 0.0$ | $-0.51 \pm 0.21$ | $0.06 \pm 0.09$ |
| | PNN [10] | $1.03 \pm 0.14$ | $8.4 \pm 0.0$ | $0.03 \pm 0.13$ | $0.00 \pm 0.00$ |
| | SAC-N | $1.00 \pm 0.15$ | $8.0 \pm 0.0$ | $0.00 \pm 0.14$ | $0.00 \pm 0.00$ |
| | FT-N | $1.16 \pm 0.20$ | $8.0 \pm 0.0$ | $0.16 \pm 0.19$ | $0.00 \pm 0.00$ |
| | CSP [5] | $\mathbf{1.27 \pm 0.27}$ | $5.4 \pm 1.3$ | $\mathbf{0.27 \pm 0.26}$ | $0.00 \pm 0.01$ |
| | Ours | $1.17 \pm 0.15$ | $2.1 \pm 0.0$ | $0.01 \pm 0.15$ | $0.01 \pm 0.01$ |

Table 6: Detailed results on 4 Ant scenarios. Baseline results are taken from [5]. New results are collected using 10 different seeds and presented with mean and standard deviation.

| | Method | Performance ↑ | Model size ↓ | Transfer ↑ | Forgetting ↓ |
|---|---|---|---|---|---|
| **Forgetting** | FT-1 | $1.31 \pm 0.33$ | $\mathbf{1.0 \pm 0.0}$ | $0.36 \pm 0.20$ | $0.05 \pm 0.23$ |
| | FT-L2 | $0.76 \pm 0.27$ | $2.0 \pm 0.0$ | $-0.24 \pm 0.24$ | $0.00 \pm 0.04$ |
| | PackNet [9] | $1.13 \pm 0.20$ | $2.0 \pm 0.0$ | $0.13 \pm 0.19$ | $0.00 \pm 0.00$ |
| | EWC [6] | $1.12 \pm 0.21$ | $3.0 \pm 0.0$ | $0.30 \pm 0.15$ | $0.17 \pm 0.22$ |
| | PNN [10] | $0.97 \pm 0.20$ | $8.0 \pm 0.0$ | $-0.03 \pm 0.19$ | $0.00 \pm 0.00$ |
| | SAC-N | $1.00 \pm 0.17$ | $8.0 \pm 0.0$ | $-0.00 \pm 0.16$ | $0.00 \pm 0.00$ |
| | FT-N | $1.36 \pm 0.26$ | $8.0 \pm 0.0$ | $\mathbf{0.36 \pm 0.25}$ | $-0.00 \pm 0.00$ |
| | CSP [5] | $1.03 \pm 0.14$ | $3.7 \pm 1.2$ | $0.03 \pm 0.13$ | $0.00 \pm 0.00$ |
| | Ours | $\mathbf{1.46 \pm 0.15}$ | $2.1 \pm 0.0$ | $0.20 \pm 0.15$ | $\mathbf{-0.00 \pm 0.00}$ |
| **Transfer** | FT-1 | $0.08 \pm 0.14$ | $\mathbf{1.0 \pm 0.0}$ | $-0.28 \pm 0.20$ | $0.64 \pm 0.15$ |
| | FT-L2 | $0.44 \pm 0.12$ | $2.0 \pm 0.0$ | $-0.44 \pm 0.07$ | $0.12 \pm 0.09$ |
| | PackNet [9] | $0.89 \pm 0.09$ | $2.0 \pm 0.0$ | $-0.11 \pm 0.09$ | $-0.00 \pm 0.00$ |
| | EWC [6] | $0.22 \pm 0.05$ | $3.0 \pm 0.0$ | $-0.78 \pm 0.04$ | $0.00 \pm 0.00$ |
| | PNN [10] | $\mathbf{1.02 \pm 0.05}$ | $8.0 \pm 0.0$ | $\mathbf{0.02 \pm 0.05}$ | $0.00 \pm 0.00$ |
| | SAC-N | $1.00 \pm 0.08$ | $8.0 \pm 0.0$ | $0.00 \pm 0.07$ | $-0.00 \pm 0.00$ |
| | FT-N | $0.83 \pm 0.12$ | $8.0 \pm 0.0$ | $-0.17 \pm 0.12$ | $-0.00 \pm 0.00$ |
| | CSP [5] | $0.93 \pm 0.10$ | $4.3 \pm 0.6$ | $-0.07 \pm 0.09$ | $\mathbf{-0.00 \pm 0.00}$ |
| | Ours | $0.76 \pm 0.07$ | $2.1 \pm 0.0$ | $-0.32 \pm 0.07$ | $0.00 \pm 0.01$ |
| **Robustness** | FT-1 | $0.34 \pm 0.06$ | $\mathbf{1.0 \pm 0.0}$ | $-0.16 \pm 0.04$ | $0.50 \pm 0.09$ |
| | FT-L2 | $0.61 \pm 0.08$ | $2.0 \pm 0.0$ | $-0.42 \pm 0.05$ | $\mathbf{-0.03 \pm 0.06}$ |
| | PackNet [9] | $0.74 \pm 0.05$ | $2.0 \pm 0.0$ | $-0.26 \pm 0.04$ | $0.00 \pm 0.00$ |
| | EWC [6] | $0.54 \pm 0.08$ | $3.0 \pm 0.0$ | $-0.47 \pm 0.07$ | $-0.01 \pm 0.02$ |
| | PNN [10] | $0.98 \pm 0.19$ | $8.0 \pm 0.0$ | $-0.02 \pm 0.18$ | $-0.00 \pm 0.00$ |
| | SAC-N | $\mathbf{1.00 \pm 0.09}$ | $8.0 \pm 0.0$ | $\mathbf{-0.00 \pm 0.09}$ | $-0.00 \pm 0.00$ |
| | FT-N | $0.80 \pm 0.09$ | $8.0 \pm 0.0$ | $-0.20 \pm 0.09$ | $-0.00 \pm 0.00$ |
| | CSP [5] | $0.60 \pm 0.11$ | $4.0 \pm 0.8$ | $-0.40 \pm 0.10$ | $0.00 \pm 0.00$ |
| | Ours | $0.73 \pm 0.11$ | $2.1 \pm 0.0$ | $-0.33 \pm 0.11$ | $-0.02 \pm 0.03$ |
| **Compositionality** | FT-1 | $0.35 \pm 0.49$ | $\mathbf{1.0 \pm 0.0}$ | $0.32 \pm 0.89$ | $0.97 \pm 0.73$ |
| | FT-L2 | $1.33 \pm 0.35$ | $2.0 \pm 0.0$ | $0.08 \pm 0.37$ | $\mathbf{-0.25 \pm 0.18}$ |
| | PackNet [9] | $1.54 \pm 0.50$ | $2.0 \pm 0.0$ | $0.54 \pm 0.47$ | $-0.00 \pm 0.00$ |
| | EWC [6] | $0.31 \pm 0.62$ | $3.0 \pm 0.0$ | $-0.07 \pm 0.47$ | $0.62 \pm 0.27$ |
| | PNN [10] | $0.95 \pm 0.81$ | $8.0 \pm 0.0$ | $-0.05 \pm 0.77$ | $-0.00 \pm 0.00$ |
| | SAC-N | $1.00 \pm 1.17$ | $8.0 \pm 0.0$ | $0.00 \pm 1.11$ | $0.00 \pm 0.00$ |
| | FT-N | $0.88 \pm 0.35$ | $8.0 \pm 0.0$ | $-0.12 \pm 0.34$ | $-0.00 \pm 0.00$ |
| | CSP [5] | $1.88 \pm 0.33$ | $3.6 \pm 0.4$ | $\mathbf{0.88 \pm 0.32}$ | $-0.00 \pm 0.01$ |
| | Ours | $\mathbf{1.95 \pm 0.11}$ | $2.1 \pm 0.0$ | $0.51 \pm 0.11$ | $-0.00 \pm 0.01$ |
| **Aggregate** | FT-1 | $0.52 \pm 0.26$ | $\mathbf{1.0 \pm 0.0}$ | $0.06 \pm 0.33$ | $0.54 \pm 0.30$ |
| | FT-L2 | $0.78 \pm 0.20$ | $2.0 \pm 0.0$ | $-0.25 \pm 0.18$ | $\mathbf{-0.04 \pm 0.09}$ |
| | PackNet [9] | $1.08 \pm 0.21$ | $2.0 \pm 0.0$ | $0.08 \pm 0.20$ | $0.00 \pm 0.00$ |
| | EWC [6] | $0.55 \pm 0.24$ | $3.0 \pm 0.0$ | $-0.26 \pm 0.18$ | $0.20 \pm 0.13$ |
| | PNN [10] | $0.98 \pm 0.31$ | $8.0 \pm 0.0$ | $-0.02 \pm 0.30$ | $0.00 \pm 0.00$ |
| | SAC-N | $1.00 \pm 0.38$ | $8.0 \pm 0.0$ | $0.00 \pm 0.36$ | $0.00 \pm 0.00$ |
| | FT-N | $0.97 \pm 0.20$ | $8.0 \pm 0.0$ | $-0.03 \pm 0.20$ | $-0.00 \pm 0.00$ |
| | CSP [5] | $1.11 \pm 0.17$ | $3.9 \pm 0.8$ | $\mathbf{0.11 \pm 0.16}$ | $0.00 \pm 0.00$ |
| | Ours | $\mathbf{1.22 \pm 0.11}$ | $2.1 \pm 0.0$ | $0.02 \pm 0.11$ | $-0.01 \pm 0.01$ |

Table 7: Detailed results on the Humanoid scenario. Baseline results are taken from [5]. New results are collected using 10 different seeds and presented with mean and standard deviation.

| Method | Performance ↑ | Model size ↓ | Transfer ↑ | Forgetting ↓ |
|---|---|---|---|---|
| FT-1 | $0.71 \pm 0.07$ | $\mathbf{1.0 \pm 0.0}$ | $0.10 \pm 0.23$ | $0.38 \pm 0.27$ |
| FT-L2 | $0.68 \pm 0.28$ | $2.0 \pm 0.0$ | $0.01 \pm 0.31$ | $0.33 \pm 0.28$ |
| PackNet [9] | $0.96 \pm 0.21$ | $2.0 \pm 0.0$ | $-0.04 \pm 0.20$ | $-0.00 \pm 0.00$ |
| EWC [6] | $0.94 \pm 0.01$ | $3.0 \pm 0.0$ | $-0.05 \pm 0.02$ | $0.01 \pm 0.02$ |
| PNN [10] | $0.98 \pm 0.26$ | $4.0 \pm 0.0$ | $-0.02 \pm 0.30$ | $0.00 \pm 0.00$ |
| SAC-N | $1.00 \pm 0.29$ | $4.0 \pm 0.0$ | $0.00 \pm 0.21$ | $-0.00 \pm 0.00$ |
| FT-N | $0.65 \pm 0.46$ | $4.0 \pm 0.0$ | $-0.35 \pm 0.35$ | $-0.00 \pm 0.00$ |
| CSP [5] | $1.76 \pm 0.19$ | $3.4 \pm 0.3$ | $\mathbf{0.75 \pm 0.16}$ | $-0.00 \pm 0.00$ |
| Ours | $\mathbf{1.78 \pm 0.22}$ | $2.0 \pm 0.0$ | $0.14 \pm 0.22$ | $\mathbf{-0.00 \pm 0.00}$ |

Table 8: Additional results of our method on Brax domains, including the mean and standard deviation obtained from 10 runs, accompanied by a 95% bootstrap confidence interval (around the mean).

| | Scenario | Performance | 95% confidence interval |
|---|---|---|---|
| Halfcheetah | Forgetting | $1.31 \pm 0.21$ | [1.11, 1.40] |
| | Transfer | $1.42 \pm 0.19$ | [1.29, 1.52] |
| | Robustness | $1.07 \pm 0.12$ | [0.98, 1.13] |
| | Compositionality | $0.88 \pm 0.09$ | [0.81, 1.92] |
| | Aggregate | $1.17 \pm 0.15$ | [1.04, 1.24] |
| Ant | Forgetting | $1.46 \pm 0.15$ | [1.36, 1.55] |
| | Transfer | $0.76 \pm 0.07$ | [0.71, 0.79] |
| | Robustness | $0.73 \pm 0.11$ | [0.68, 0.81] |
| | Compositionality | $1.95 \pm 0.11$ | [1.87, 2.00] |
| | Aggregate | $1.22 \pm 0.11$ | [1.15, 1.29] |
| | Humanoid | $1.78 \pm 0.22$ | [1.65, 1.92] |

Table 9: Detailed success rates (↑) on 8 triplet (CW3) scenarios from Continual World. * indicates results taken from [5]. The rest of the results are collected from 3 different seeds and presented with mean and standard deviation. Aggregated results are shown in the main paper.

| Method | T1 | T2 | T3 | T4 |
|---|---|---|---|---|
| FT-1* | $0.24 \pm 0.13$ | $0.25 \pm 0.07$ | $0.39 \pm 0.16$ | $0.34 \pm 0.05$ |
| FT-L2 | $0.21 \pm 0.15$ | $0.21 \pm 0.06$ | $0.33 \pm 0.19$ | $0.31 \pm 0.06$ |
| PackNet [9] | $0.62 \pm 0.21$ | $0.58 \pm 0.17$ | $0.80 \pm 0.11$ | $0.41 \pm 0.07$ |
| EWC [6]* | $0.45 \pm 0.12$ | $0.27 \pm 0.09$ | $0.38 \pm 0.09$ | $0.31 \pm 0.12$ |
| PNN [10]* | $\mathbf{0.84 \pm 0.08}$ | $0.72 \pm 0.17$ | $\mathbf{0.90 \pm 0.05}$ | $0.43 \pm 0.08$ |
| SAC-N* | $0.69 \pm 0.17$ | $0.71 \pm 0.13$ | $0.79 \pm 0.19$ | $0.47 \pm 0.14$ |
| FT-N* | $0.77 \pm 0.08$ | $\mathbf{0.86 \pm 0.10}$ | $0.78 \pm 0.15$ | $0.49 \pm 0.14$ |
| CSP [5]* | $0.76 \pm 0.20$ | $0.79 \pm 0.03$ | $0.82 \pm 0.08$ | $\mathbf{0.58 \pm 0.09}$ |
| Ours | $0.79 \pm 0.12$ | $0.80 \pm 0.09$ | $0.83 \pm 0.11$ | $0.56 \pm 0.08$ |

| Method | T5 | T6 | T7 | T8 |
|---|---|---|---|---|
| FT-1* | $0.30 \pm 0.01$ | $0.32 \pm 0.25$ | $0.17 \pm 0.07$ | $0.34 \pm 0.05$ |
| FT-L2 | $0.21 \pm 0.16$ | $0.11 \pm 0.04$ | $0.14 \pm 0.06$ | $0.20 \pm 0.14$ |
| PackNet [9] | $0.34 \pm 0.10$ | $0.34 \pm 0.02$ | $0.36 \pm 0.15$ | $0.47 \pm 0.11$ |
| EWC [6]* | $0.32 \pm 0.07$ | $0.33 \pm 0.18$ | $0.20 \pm 0.10$ | $0.32 \pm 0.08$ |
| PNN [10]* | $0.33 \pm 0.23$ | $0.46 \pm 0.21$ | $0.44 \pm 0.12$ | $0.36 \pm 0.20$ |
| SAC-N* | $0.60 \pm 0.13$ | $0.55 \pm 0.11$ | $0.54 \pm 0.15$ | $0.45 \pm 0.12$ |
| FT-N* | $0.52 \pm 0.13$ | $0.61 \pm 0.06$ | $\mathbf{0.61 \pm 0.13}$ | $0.52 \pm 0.06$ |
| CSP [5]* | $0.58 \pm 0.06$ | $0.54 \pm 0.06$ | $0.58 \pm 0.04$ | $0.53 \pm 0.08$ |
| Ours | $\mathbf{0.62 \pm 0.13}$ | $\mathbf{0.61 \pm 0.06}$ | $0.57 \pm 0.07$ | $\mathbf{0.54 \pm 0.07}$ |

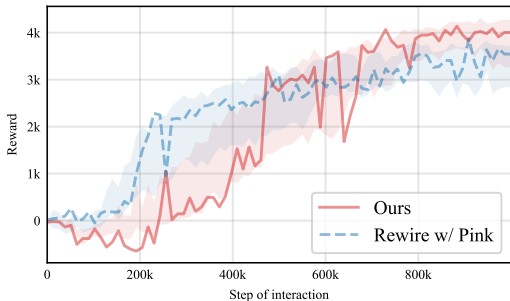

Figure 1: Effectiveness of multi-mode strategy in the first stage, compared to pink noise [3]. The curves depict the median, with shaded areas showing 95% bootstrap confidence interval for the mean.

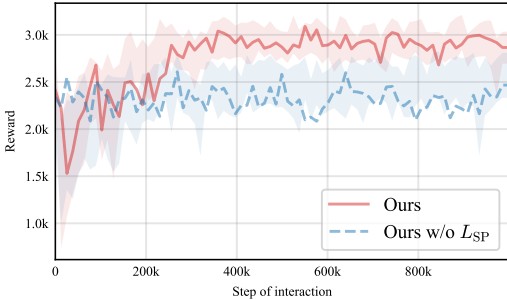

Figure 2: Effectiveness of alignment loss $L_{SP}$ in the second stage. The detailed setups follow Fig. 1.

Table 10: Compairson of multi-mode strategy with another ensemble method, BatchEnsemble [12].

| Method | Performance | 95% confidence interval | Model size |
|---|---|---|---|
| BatchEnsemble [12] | $0.94 \pm 0.23$ | [0.81, 1.08] | **1.1** |
| Ours | $\mathbf{1.31 \pm 0.21}$ | [**1.11**, **1.40**] | 2.1 |

Table 11: Comparison to CSP [5] at similar model sizes. CSP-S reduces the network width to 175, while Ours-L expands it to 384. See Fig. 4b in the main paper for a more intuitive visualization.

| Method | Performance | 95% confidence interval | Model size |
|---|---|---|---|
| CSP-S | $1.27 \pm 0.15$ | [1.14, 1.32] | 2.3 |
| Ours | $1.31 \pm 0.21$ | [1.11, 1.40] | **2.1** |
| CSP [5] | $\mathbf{1.41 \pm 0.07}$ | - | 4.5 |
| Ours-L | $1.38 \pm 0.10$ | [**1.31**, **1.42**] | 4.6 |

## B.3 Ablation studies

This section provides additional justification for our proposed rewiring designs. First, to demonstrate the exploration efficacy of our multi-mode strategy, we compare it against an existing method called pink noise [3]. As shown in Fig. 1, while the single-mode baseline with pink noise exhibits rapid initial learning, its performance plateaus over time. In contrast, our full method with multi-mode strategy effectively avoids this suboptimal situation and achieves the highest final performance.

To validate the effectiveness of the proposed alignment mechanism, we plot the performance curves in Fig. 2 (truncated to the second learning stage), where the full model with alignment mechanism exhibits the fastest adaptation and highest final performance compared to other variants. This also leads to better results than alternative ensemble methods such as BatchEnsemble [12] in Table 10.

Lastly, to examine the scalability of our approach, we compare it to CSP at similar model sizes. Table 11 show that our method achieves slightly higher mean performance than CSP-S at small sizes, while delivering a noticeable improvement and closing the gap with CSP when scaling up.