# OpenReview forum: "Rewiring Neurons in Non-Stationary Environments"
_NeurIPS.cc/2023/Conference — NeurIPS 2023 spotlight_

### Official Review · Reviewer_NgVj · 2023-06-16

**Soundness:** 2 fair
**Presentation:** 3 good
**Contribution:** 2 fair
**Rating:** 5
**Confidence:** 3

**Summary:**

The paper presents a bio-inspired rewiring technique to improve deep reinforcement learning (DRL), especially in continual learning in non-stationary environment. The rewiring is implemented using permutation matrix P for all the hidden layers in MLP. There are several benefits. First, by using a set of different wirings, the agent executes more diverse policy which is claimed to improve exploration. Second, rewiring with a novel regularization trick can alleviate the stability-plasticity dilemma in continual learning. The effectiveness of the second claim is proven by robotic control experiments. The proposed rewiring method achieves SOTA with less network parameters. Ablation studies are also presented for better understanding the technique.

---- post rebuttal -----

I appreciate the authors' reply. I lead toward acceptance.

**Strengths:**

- The writing is clear and easy to follow
- The idea is interesting and well motivated
- The proposed method is well evaluated with diverse tasks and ablation studies
- Connections with neuroscience is well discussed

**Weaknesses:**

- The claim in 3.3 lack supporting evidences. Although ablation studies show a performance degragation without multi-mode, it is unclear how it relates to "exploration". There are various ways of improving exploration, e.g., simply using pink noise for action (https://openreview.net/forum?id=hQ9V5QN27eS). If the author would like to claim their methods' advance on exploration, comparison with other methods should be performed. And there should be empirical or theoretical evidence to show such an advantage in terms of "enable the agents to explore unseen environment (line 168)" rather than Fig.3 or overall performance.

- The proposed method achives similar performance (I would say no statistically significant difference from CSP) while using less parameters, which is a good thing. However, since the network is small (a few FC layers of width 256),  such an advantage is not obviously important. I believe it would be much more exciting if the authors can show that (1) rewiring largely outperforms CSP with simular model size (could be very small network that can work onsingle chip microcomputer) or (2) performs similar or better than CSP with less energy consumption in tasks that require heavy computation resource.

- Although MLP is a foundamental neural network structure, it is unclear wether the rewiring technique works well for Transformers, CNNs, RNNs, etc.

**Questions:**

- What's the computational cost and time consumption of your method comparing to others despite having less model parameters?
- Is your method compatible with evolutionary algorithms if using non-differentiable sorting for P?
- What's the scalability of the rewiring technique? More specifically, will the performance improve if you make your model larger (comparable with CSP in Table 1)?

**Limitations:**

The authors have discussed limitation reasonably in the manuscript.

---

> ### Author Rebuttal · Authors · 2023-08-09
>
> Thank you for the very constructive comments. Our responses are provided below.
>
> [W1] Justification for multi-mode strategy
>
> * In Figure 2a of the global response, we compare the exploration efficacy of our multi-mode strategy against pink noise [1]. While the single-mode baseline with pink noise exhibits rapid initial learning, its performance plateaus over time. In contrast, our full method with multi-mode strategy effectively avoids this suboptimal situation and achieves the highest final performance.
> * In addition, we recommend that the reviewer refer to Figure 5 and line 299 in the main paper, where a similar empirical justification for multi-mode rewiring is provided.
>
> [W2] Additional comparison with CSP
>
> * To compare at "similar model sizes", we experiment with CSP using a reduced network width of 175 (denoted as CSP-S). This aligns the model size of the two methods on the HalfCheetah/forgetting scenario. As shown in the results below, our method achieves slightly higher mean performance than CSP-S at similar small sizes.
>
>   |       | H/F performance | 95% confidence interval       | Model size |
>   | ----- | --------------- | ------------ | ---------- |
>   | CSP-S | 1.27 $\pm$ 0.15 | [1.14, 1.32] | 2.3        |
>   | Ours  | 1.31 $\pm$ 0.21 | [1.11, 1.40] | 2.1        |
>
> [W3] Exploration of network architectures
>
> * Currently, our work is focused on MLP due to its standard use in the continual RL benchmarks, and the observations we've made already provide important insights into the field. Meanwhile, we recognize the value of exploring rewiring techniques in other architectures such as Transformers and CNNs, possibly moving on to datasets like Split ImageNet. We will continue to investigate bio-inspired approaches like rewiring with more sophisticated architectures and tasks in the future.
>
> [Q1] Computational efficiency
>
> * In Appendix A.4, we have shown that our method outperforms CSP in terms of time efficiency. Here is another comparison using multiply-add operations (MACs) on the HalfCheetah/forgetting scenario:
>
>   |      | MACs (M) | Model size |
>   | ---- | -------- | ---------- |
>   | FT-1 | 0.14     | 1.0        |
>   | PNN  | 1.08     | 8.0        |
>   | CSP  | 0.63     | 4.5        |
>   | Ours | 0.48     | 2.1        |
>
>   As can be seen, our method has a lower computational cost than both PNN and CSP, despite the need for two forward passes (to compute the distillation loss $L_{\textrm{KL}}$ between wirings).
>
> [Q2] Compatibility with evolutionary algorithms
>
> * Thank you for bringing attention to this line of work [2,3]. Integrating evolutionary algorithms into our model appears to be difficult, due to the considerable engineering effort required for their fitness functions. And the efficiency may not be comparable to differentiable sorting. Nevertheless, evolutionary algorithms represent a more biologically plausible solution, since there is no clear evidence that the brain implements differentiable algorithms, so they are still worth exploring.
>
> [Q3] Scalability
>
> * We validate the scalability of our method by expanding the network width to 384 (denoted as Ours-L), which allows a direct comparison with the original CSP results as follows:
>
>   |        | H/F performance | 95% confidence interval       | Model size |
>   | ------ | --------------- | ------------ | ---------- |
>   | Ours   | 1.31 $\pm$ 0.21 | [1.11, 1.40] | 2.1        |
>   | Ours-L | 1.38 $\pm$ 0.10 | [1.31, 1.42] | 4.6        |
>   | CSP    | 1.41 $\pm$ 0.07 | -            | 4.5        |
>
>   The comparison shows that Ours-L has a noticeable performance improvement, closing the gap with CSP. For a more intuitive illustration, please see Figure 1b in the global response.
>
> References
>
> [1] Eberhard, O., Hollenstein, J., Pinneri, C., & Martius, G. (2023). Pink noise is all you need: Colored noise exploration in deep reinforcement learning. In *International Conference on Learning Representations*.
>
> [2] Scharnow, J., Tinnefeld, K., & Wegener, I. (2005). The analysis of evolutionary algorithms on sorting and shortest paths problems. *Journal of Mathematical Modelling and Algorithms*, *3*, 349-366.
>
> [3] Bassin, A., & Buzdalov, M. (2020). The (1+($\lambda$, $\lambda$)) genetic algorithm for permutations. In *Proceedings of the 2020 Genetic and Evolutionary Computation Conference Companion* (pp. 1669-1677).

---

> > ### Comment · Reviewer_NgVj · 2023-08-12
> >
> > Thanks for replying and conducting additional experiments.  However, there still lack direct evidence of "multi-mode enables the agents to explore unseen environment". The final performance is increased, In Figure 2a of the global response, it shows using pink noise . According to my experience in deep RL, better exploration means faster learning at the begining since it can collect more diverse experience quicker. But multi-mode shows a slower learning than pink-noise at the begining, which makes me concern that the performance gain is not due to better exploration but other reasons.

---

> > > ### Author Response · Authors · 2023-08-13
> > >
> > > Thank you so much for your response, we would like to address your concerns as follows.
> > >
> > > * We assume that "direct evidence" of exploration refers to visualization of state space coverage, like the 2D trajectories in [1, 2]. However, this is not feasible given the high-dimensional spaces of continual RL tasks. As an alternative, we use policy diversity (Figure 3) and performance evolution (Figure 5) to demonstrate exploration. For a more rigorous description, line 168 will be revised to "multi-mode rewiring enables the agent to explore various policies".
> > > * Our results resonate with the well-known exploration-exploitation tradeoff: exploration may yield lower short-term rewards, but higher long-term rewards [3]. The reviewer commented that "better exploration means faster learning at the beginning since it can collect more diverse experience quicker". However, several deep RL results, such as Figure 7 (left) in [2], Figure 5 (right) in [4], and Figures 4 (left) & 5 (left) in [5], also support that exploration can exhibit slower learning at the beginning.
> > > * Lastly, it is worth noting that our rewiring method is beneficial even when faster learning is prioritized, as the rewiring-based methods (Rewire and Ours) consistently outpace the non-rewiring baseline (FT) in Figure 5.
> > >
> > > References:
> > >
> > > [1] Houthooft, R., Chen, X., Duan, Y., Schulman, J., De Turck, F., & Abbeel, P. (2016). VIME: Variational information maximizing exploration. *Advances in Neural Information Processing Systems*, *29*, 1117-1125.
> > >
> > > [2] Eysenbach, B., Gupta, A., Ibarz, J., & Levine, S. (2019). Diversity is all you need: Learning skills without a reward function. In *International Conference on Learning Representations*.
> > >
> > > [3] Sutton, R. S., & Barto, A. G. (2018). *Reinforcement Learning: An Introduction*. MIT press.
> > >
> > > [4] Pathak, D., Gandhi, D., & Gupta, A. (2019). Self-supervised exploration via disagreement. In *International Conference on Machine Learning* (pp. 5062-5071).
> > >
> > > [5] Badia, A. P., Sprechmann, P., Vitvitskyi, A., Guo, D., Piot, B., Kapturowski, S., ... & Blundell, C. (2020). Never give up: Learning directed exploration strategies. In *International Conference on Learning Representations*.

---

> > > > ### Comment · Reviewer_NgVj · 2023-08-14
> > > >
> > > > Thanks for the clarifications, which address some of my concerns. I raise my score to 5 accodingly. Nonetheless, I believe it remains to be investigated the detailed mechanism by which multi-mode policies facilitates learning, and clarification of how it relates to policy ensemble such as [a].
> > > >
> > > > [a] Lee, Kimin, et al. "Sunrise: A simple unified framework for ensemble learning in deep reinforcement learning." International Conference on Machine Learning. PMLR, 2021.

---

### Official Review · Reviewer_nWMB · 2023-06-29

**Soundness:** 4 excellent
**Presentation:** 3 good
**Contribution:** 4 excellent
**Rating:** 8
**Confidence:** 4

**Summary:**

The authors propose a means to efficiently expand the capacity of a neural network, namely connection permutations. The approach interleaves permutations matrices between layers of a neural network, such that input-output relationships can be adapted during learning in addition to learning the weight matrices. The authors focus in particular on RL in non-stationary environments, where adapting to new tasks can result in catastrophic forgetting of older tasks.

The permutations are learned with an existing differentiable approximation to the argsort operator. In addition, the authors propose to (a)  allow the agent to cache permutations from previous tasks; (b) have the agent sample from multiple permutations to encourage exploration; and (c) align cached permutations with the latest weight matrices to ensure that previously learned policies are still consistent with the latest network, preventing catastrophic forgetting. Evaluating on a collection of continual RL tasks, the authors show SOTA or near-SOTA performance while using significantly fewer parameters than other techniques.

**Strengths:**

Originality:
The argsort relaxation is an existing method, and differentiable permutations have been used for sorting inputs and in the attention maps of transformers. The authors also point out that neuron permutations have previously been used to explore network alignment. However, a permutation-based approach to network topology does not appear to have been exploited to manage the stability-plasticity tradeoff. This is a novel insight and a meaningful step beyond the existing literature, to the best of my knowledge.

Clarity:
Overall I found the presentation of the main idea to be clear, with helpful diagrams, although there were some issues on specific points (see below).

Significance:
The application of rewiring alongside weight learning has potential impact beyond non-stationary RL, including in other continual learning tasks, RL with exploration and either RL or non-RL tasks comprised of modular sub-tasks. For this reason, the proposed method is a valuable contribution to the literature.

**Weaknesses:**

Eq. 11 appears to require caching the previous state of the weights as well as the permutations, which could be a significant memory overhead for large networks.

It's not clear to me, but it appears that the weights are cached on presenting a new task in a rule-based way; i.e. an external signal to the network indicating that the agent has started a new task. Does this mean that the model technically receives more information than the baselines, in the form of a signal to cache?

One clarity issue was in the computational neuroscience vs. ML direction of the paper. Based on the title in particular, I expected the focus to be on biologically plausible non-stationary RL / navigation, and it took me well into the Related Work to realize that it was primarily an ML paper. I would probably include a reference to permutation in the the title if you want it to be picked up by the ML and RL communities (e.g. "Permuting Neurons in Non-Stationary Reinforcement Learning"). On the other hand, I think the title is fine if the authors primarily expect to go down the comp neuro / bio plausible route with this work as suggested in the Conclusions.

I found the definitions of the ablations to be somewhat confusing in part because they were presented differently in the main text than the paper. The paper starts by describing rewiring and multi-modality together in 3.3, then caching and alignment together in 3.4, while the ablation table has rewiring, then caching, then multi-modality, then alignment. Can you include paragraph sub-headers in the description for each of these interventions in 3.3 and 3.4 to make it easier for the reader to turn back and match up with the ablation table?

Can you re-state what K and (especially) alpha and beta correspond to in the subtitle of Figure 6?

Overall I think the paper is strong. A higher rating (9/10) would likely require a more theoretically rigorous analysis of the impact of permutations on network capacity, a demonstration of lift on tasks outside of RL / non-stationary RL, or both.

**Questions:**

- See the point above about whether the model indirectly receives information about when to cache the weights. If so, are there means (e.g. information-theoretic surprise) by which the model could detect a task change and initiate weight caching itself?

**Limitations:**

The authors include a short assessment of Limitations in the Conclusions. I don't foresee any significant negative societal impact.

---

> ### Author Rebuttal · Authors · 2023-08-09
>
> Thank you for the very constructive comments. We would clarify as follows.
>
> [W1] Memory overhead
>
> * Our method requires caching one previous weight $\boldsymbol{W}^{t-1}$ and all previous permutations. Among them, the weight accounts for most of the overhead, while the permutations prove to be highly parameter-efficient (see lines 142 and 186). Overall, the memory cost is only slightly more than twice the base size, remaining competitive with other existing methods.
>
> [W2] Issue with task identifiers
>
> * The model indeed receives an "external signal" indicating the start of a new task, known as the task identifier. However, it is commonly used by the compared methods. For instance, PNN and CSP expand the model upon each new task, while EWC computes the Fisher information matrix at the end of each task. Therefore, our method does not acquire more information than these baselines.
>
> [W3] Direction in computational neuroscience vs. ML
>
> * This paper aims to approach an ML problem from a bio-inspired route, with "Rewiring Neurons" in the title underscoring our neuroscience motivation. We've touched on some related discussion in Section 3.5 and the Conclusion, and will continue to revise the paper to make our intent clearer.
>
> [W4] Presentation of the ablation table.
>
> * We apologize for the different order of the ablation table, as it goes from rewiring (Section 3.2), then caching (Section 3.4), then multi-mode (Section 3.3), then alignment (Section 3.4). To help readers locate each design in the main text, we will add two paragraph sub-headers "Caching each wiring" and "Aligning wirings with weights" in Section 3.4.
>
> [W5] Hyperparameters in Figure 6
>
> * $K$ is the number of modes in multi-mode rewiring (Eq. 5), $\alpha$ is the coefficient for the distillation loss $L_{\textrm{KL}}$ (Eq. 6), and $\beta$ is the coefficient for the regularization loss $L_{\textrm{SP}}$ (Eq. 11). For clarification, we will re-state the definition of $\alpha$ and $\beta$ (originally in line 218) in Figure 6.
>
> [W6-1] Impact of permutation on network capacity
>
> * The permutation layers do not affect the network capacity. Instead, they facilitate exploration of the existing weight space (by enabling traversal from one weight point to another via permutation transforms). This largely enhances the network's adaptivity, as shown in the continual RL experiments.
>
> [W6-2] Demonstration on non-RL tasks
>
> * We have not yet conducted such experiments, but looking beyond RL tasks presents intriguing challenges. Take visual tasks as an example: neuroscience research [1] indicates that synapses in the visual cortex exhibit much higher stability. This could imply a need for new rewiring approaches with more moderate rewiring intensity.
>
> [Q1] Detecting task changes
>
> * This has been investigated in the field of task-free continual learning [2] where task identifiers are absent. For example, Rao et al. [3] suggested maintaining a buffer for poorly modeled samples and then expanding the model when the buffer reaches a critical size. Ardywibowo et al. [4] proposed to detect task changes using an energy-based novelty score.
>
> References
>
> [1] Grutzendler, J., Kasthuri, N., & Gan, W. B. (2002). Long-term dendritic spine stability in the adult cortex. *Nature*, *420*(6917), 812-816.
>
> [2] Aljundi, R., Kelchtermans, K., & Tuytelaars, T. (2019). Task-free continual learning. In *Proceedings of the IEEE/CVF Conference on Computer Vision and Pattern Recognition* (pp. 11254-11263).
>
> [3] Rao, D., Visin, F., Rusu, A., Pascanu, R., Teh, Y. W., & Hadsell, R. (2019). Continual unsupervised representation learning. *Advances in neural information processing systems*, *32*, 7647-7657.
>
> [4] Ardywibowo, R., Huo, Z., Wang, Z., Mortazavi, B. J., Huang, S., & Qian, X. (2022). VariGrow: Variational architecture growing for task-agnostic continual learning based on Bayesian novelty. In *International Conference on Machine Learning* (pp. 865-877).

---

> > ### Comment · Reviewer_nWMB · 2023-08-14
> > **Re: Rebuttal**
> >
> > Thank you to the authors for their response and improvements to clarity. I maintain my current score and recommendation of acceptance.

---

### Official Review · Reviewer_AnPE · 2023-07-02

**Soundness:** 3 good
**Presentation:** 3 good
**Contribution:** 3 good
**Rating:** 7
**Confidence:** 3

**Summary:**

The authors propose a new architectural method for continual reinforcement learning. The method relies on training not just the weights of the network, but also the arrangement of the neurons in each layer (implemented as permutation vectors based on a learned score vector for each layer of neurons).

The method continually adjusts previously saved permutations/rewirings, by ensuring that they would produce similar outputs when applied to the new weights, despite constant changes in the actual weights (IIUC).

In its current form, the method requires task IDs, but no replay buffers.

Various experiments suggest that the method is competitive with other established methods with similar requirements, despite its conceptual simplicity.



**Strengths:**

- The method is novel (to my knowledge) and elegant.

- The results seem interesting.

- The exposition is (mostly) clear

**Weaknesses:**

I don't see any major weakness, except for the limitations noted by the authors in the Conclusion. I would appreciate some clarifications, as detailed below.

**Questions:**

- Eq. 3: While I understand NumPy notation, some readers might not. Please explain it a bit more.

- Eq. 4: This seems to be the most important equation, but it is difficult to understand. What is the d() function? Apparently it is not the dimensionality of the layers, which is also (confusingly) called d in the next paragraph? Please clarify this.

- Figure 3 shows the divergence and convergence between the various policies, but not their performance. It would be useful to have an additional curve indicating (say) median performance.

- Section 3.4: IIUC, the method here is to continuously adjust the previously found wirings P(t-k), so that they will produce similar outputs as they did with their original weights w(t-k), when applied to the new weights w(t), despite the fact that the weights w(t) keep changing (if so, a one-sentence explanation of this type would be helpful).

- It seems that you are learning W(t) and P’/P’’ together, in parallel, through the loss in equation 11. An alternative would be to simply learn the new W(t) in isolation, then afterwards compute the P’/P’’ “offline”, by applying Eq 11 with frozen W(t) (since Eq. 11 doesn’t seem to require any interaction with the environment). Why not do the latter?

- Can you please confirm whether you simply train one new rewiring (or more precisely one new P’/P’’ pair) per task? That is, does the ’t’ index on Pt/Wt correspond to the tasks? Or is there something else involved?






**Limitations:**

The authors note some limitations of their approach in the Conclusion. These do not seem deal-breaking to me, since many of these limitations are shared by other approaches requiring task IDs.

---

> ### Author Rebuttal · Authors · 2023-08-09
>
> Thank you for providing valuable feedback. Our responses to your questions are below.
>
> [Q1] NumPy notation in Eq. 3
>
> * Thanks for pointing out the NumPy notation in $\boldsymbol{I}[\boldsymbol{z}_l,:]$. It rearranges the rows of the identity matrix $\boldsymbol{I}$ according to the indices $\boldsymbol{z}_l$, resulting in a permutation matrix. For example, consider $\boldsymbol{v}_l=\begin{pmatrix}0&4&2\end{pmatrix}^\top$ and $\boldsymbol{z}_l=\mathrm{argsort}(\boldsymbol{v}_l)=\begin{pmatrix}0&2&1\end{pmatrix}^\top$, then we have:
>   $$
>   \begin{aligned}
>   \boldsymbol{P}_l&=\boldsymbol{I}[\boldsymbol{z}_l,:]\\\\
>   &=\begin{pmatrix}1&0&0\\\\0&0&1\\\\0&1&0\end{pmatrix}.
>   \end{aligned}
>   $$
>
> [Q2] The $d()$ function in Eq. 4
>
> * $d()$ is a differentiable almost everywhere, semi-metric function (such as $L_1$ distance, $L_2$ distance) applied pointwise [1]. In this paper we use the $L_1$ distance. Below is a demonstration following the previous example, with $\tau=1$:
>   $$
>   \begin{aligned}
>   \hat{\boldsymbol{P}}_l&=\mathrm{softmax}\left(\frac{-d(\mathrm{sort}(\boldsymbol{v}_l)\boldsymbol{1}^\top,\boldsymbol{1}\boldsymbol{v}_l^\top)}{\tau}\right)\\\\
>   &=\mathrm{softmax}\left(-d(\begin{pmatrix}0&0&0\\\\2&2&2\\\\4&4&4\end{pmatrix},\begin{pmatrix}0&4&2\\\\0&4&2\\\\0&4&2\end{pmatrix})\right)\\\\
>   &=\mathrm{softmax}\begin{pmatrix}0&-4&-2\\\\-2&-2&0\\\\-4&0&-2\end{pmatrix}\\\\
>   &\approx\begin{pmatrix}0.867&0.016&0.117\\\\0.107&0.107&0.787\\\\0.016&0.867&0.117\end{pmatrix}.
>   \end{aligned}
>   $$
>   The result is a continuous relaxation of the permutation matrix $\boldsymbol{P}_l$, which allows end-to-end learning.
>
> * As for the naming confusion, we will rename the dimensionality of the layers to $n$ to avoid any ambiguity.
>
> [Q3] Performance of multi-mode policies
>
> * Following your suggestion, we've included performance curves in the global response (Figure 2). Figure 2a demonstrates that, despite the slower initial learning of multi-mode policies from policy divergence, they ultimately outperform upon convergence. Figure 2b shows that incorporating explicit convergence guidance improves both mean and median rewards, verifying the impact of convergence on performance.
>
> [Q4] Interpretation of Section 3.4
>
> * Your understanding is correct. Our method continuously adjusts the cached wirings to counteract weight changes, thereby improving stability. We will add a brief explanation like this in Section 3.4 for clarification.
>
> [Q5] Computing $\boldsymbol{P}'$/$\boldsymbol{P}''$ afterwards
>
> * In this case, the learning objective for $\boldsymbol{W}^t$ (Eq. 11) is computed using outdated $\boldsymbol{P}'$/$\boldsymbol{P}''$ (from the last round). Therefore, its performance is likely to be inferior to joint learning, for which we stayed with the original implementation.
>
> [Q6] Training details
>
> * We train one new wiring $\boldsymbol{P}^t$ and one new $\boldsymbol{P}'$/$\boldsymbol{P}''$ pair per task, where the index $t$ on $\boldsymbol{P}^t$ represents the current task. The rest of the wirings ($\boldsymbol{P}^{t-k},k>0$) remain frozen and are only updated at the end of each task, so no additional training cost is involved.
>
> Reference:
>
> [1] Prillo, S., & Eisenschlos, J. (2020). SoftSort: A continuous relaxation for the argsort operator. In *International Conference on Machine Learning* (pp. 7793-7802).

---

> > ### Comment · Reviewer_AnPE · 2023-08-12
> > **Thank you**
> >
> > I appreciate the author's clarifications in response to this and other reviews. I maintain my recommendation for acceptance.

---

### Official Review · Reviewer_cvgK · 2023-07-04

**Soundness:** 3 good
**Presentation:** 2 fair
**Contribution:** 3 good
**Rating:** 6
**Confidence:** 4

**Summary:**

The paper proposes a new method for continual reinforcement learning. The idea is to leverage a neuron rewiring mechanism implemented as additional permutations of neurons between the NN weights. In the proposed algorithm, several such permutation sets are maintained, which correspond to different policies that can be used. Having those multiple policies helps to maintain exploration. As a regularization mechanism, a variant of L2 is used for combined NN weight matrices and permutation matrices. Experimental evaluation shows the strong performance of the proposed method, which often matches or outperforms baseline approaches while limiting parameter overhead.

**Strengths:**

[S1] Presented solution is quite simple and elegant. Neuron permutations provide both an effective parametrization of the NN and the mechanism for having cheap ensembles.

[S2] Experimental results are compelling, the presented approach is demonstrated to match or beat previous approaches and limit the parameter overhead.

[S3] Ablation experiments and analyses are provided, providing further insights into the method.


**Weaknesses:**

[W1] Although the paper is mostly easy to read, some sections would benefit from editing. Especially Section 3.4 seems wordy and sometimes contains unnecessary or misleading statements, e.g. line 195: “In this situation, one may interpret…” - I have to say I don’t understand this sentence. See also the Questions section of the review for more (mostly minor) suggestions.


**Questions:**

[Q1] Are rewiring mechanisms used for actor only, or both actor and critic? Have you tried different options here?

[Q2] Suggestion for an additional experiment (optional): one can see your approach as a way of maintaining a cheap set of ensemble models, each starting from one base model and modifying it with some small adapting module. It would be interesting to compare performance to similar solutions (not previously used in continual RL AFAIK), e.g. https://arxiv.org/abs/2002.06715

Some additional (minor) suggestions:

[Q3] Figure 5 - it is not super clear what is “Rewire” here.

[Q4] Figure 4 - not very easy to read (e.g. because of 4 different triangle shapes) - maybe colors would be easier to read?

[Q5] - “Validation step” label in some of the figures - a nicer alternative would be something more interpretable like training steps.


**Limitations:**

IMO authors sufficiently covered the limitations in a dedicated paragraph in the last section.

---

> ### Author Rebuttal · Authors · 2023-08-09
>
> We deeply appreciate your valuable suggestions, and we would like to address your main concerns as follows:
>
> [W1] Misleading statements in Section 3.4
>
> * We apologize for any confusion around Section 3.4. Specifically, in line 195, our aim was to present a new interpretation of catastrophic forgetting: Instead of framing it as a shift in weight space (as in EWC, SI [1], etc.), we view it as the misalignment between updated weights and fixed wiring. This perspective led to our proposal to align the wiring with the weights. We will further revise this section to improve its clarity and conciseness.
>
> [Q1] Rewiring details
>
> * The rewiring mechanism is only used for the actor. We have not rewired the critics because many baselines (such as PackNet and PNN) in Brax [2] and Continual World [3] default to standard critics. Rewiring the twin critics would impose significantly higher computational costs than those baselines.
>
> [Q2] Comparison with ensemble methods
>
> * Following your suggestion, we implement BatchEnsemble [4] in continual RL, which learns rank-one "fast weights" for adapting to each new task. The results on the HalfCheetah/forgetting scenario are as follows:
>
>   |               | H/F performance | 95% confidence interval       | Model size |
>   | ------------- | --------------- | ------------ | ---------- |
>   | BatchEnsemble | 0.94 $\pm$ 0.23 | [0.81, 1.08] | 1.1        |
>   | Ours          | 1.31 $\pm$ 0.21 | [1.11, 1.40] | 2.1        |
>
>   While BatchEnsemble is very efficient, its performance is limited by the expressiveness of rank-one matrices (the main weights are frozen during learning). Our method, on the other hand, jointly learns the weights and the wirings under an alignment scheme (Eq. 11), thus achieving better results.
>
> [Q3] "Rewire" in Figure 5
>
> * "Rewire" represents rewiring (Section 3.2) but without the multi-mode strategy (Section 3.3). We have revised the figure caption to provide a brief explanation. Please refer to Figure 2 in the global response.
>
> [Q4] Readability of markers in Figure 4
>
> * Thank you for the valuable suggestion. We've now added colors to differentiate the various triangle markers, and the updated version is included as Figure 1a in the global response.
>
> [Q5] More interpretable x label in Figures 3 and 5
>
> * We have replaced "Validation step" with "Step of interaction" in the global response. The "training steps" mentioned in the reviewing comment can be easily inferred from this. For example, on the HalfCheetah/forgetting scenario, the actor is updated every two steps of interaction, while the critics are updated every step.
>
> References
>
> [1] Zenke, F., Poole, B., & Ganguli, S. (2017). Continual learning through synaptic intelligence. In *International conference on machine learning* (pp. 3987-3995).
>
> [2] Gaya, J. B., Doan, T., Caccia, L., Soulier, L., Denoyer, L., & Raileanu, R. (2023). Building a subspace of policies for scalable continual learning. In *International Conference on Learning Representations*.
>
> [3] Wołczyk, M., Zając, M., Pascanu, R., Kuciński, Ł., & Miłoś, P. (2021). Continual world: A robotic benchmark for continual reinforcement learning. *Advances in Neural Information Processing Systems*, *34*, 28496-28510.
>
> [4] Wen, Y., Tran, D., & Ba, J. (2020). BatchEnsemble: an alternative approach to efficient ensemble and lifelong learning. In *International Conference on Learning Representations*.

---

> > ### Comment · Reviewer_cvgK · 2023-08-14
> >
> > I thank the authors for their detailed answers to my review. I choose to keep my original score, as it still reflects my evaluation of the work.

---

### Official Review · Reviewer_CtSV · 2023-07-07

**Soundness:** 1 poor
**Presentation:** 2 fair
**Contribution:** 3 good
**Rating:** 6
**Confidence:** 3

**Summary:**

This work studies the continual reinforcement learning setting. This paper proposes a method to permute the neurons in the network, and these permutations allow the exploration of a large part of the weight space. The proposed method caches weights from the prior tasks, which helps to mitigate forgetting. An additional rewiring strategy is proposed to encourage exploration when the task changes.

**Strengths:**

The proposed method is memory efficient, which is essential for lifelong learning systems as they might encounter an arbitrarily long sequence of tasks. The memory complexity is only O(k*d), where d is the number of neurons in the network and k is the number of tasks.

I like the focus on ensuring continual exploration in new environments. This issue is generally overlooked in the literature, and the emphasis on continual exploration strengthens this work.

The empirical evaluation is performed on a wide range of environments. This evaluation can provide a detailed assessment of the proposed method.

**Weaknesses:**

Although the ideas presented in this paper are interesting, the empirical evaluation could be more rigorous.
All the experiments are performed with just three random seeds (line 259), meaning the results presented in the paper are not statistically significant.

**Questions:**

The paper only considers a setting where the agent cannot access replay buffers from the previous tasks. But this seems like an arbitrary choice. If the agent has a replay, why can it not store experience from previous tasks? Can you please elaborate on the rationale for not allowing methods to store experience from previous tasks?

**Limitations:**

The empirical evaluation is this work is weak. I recommend a new paper by Patterson et al. (2023) to understand how to perform proper empirical analysis in RL.

I will consider raising my score by 3 points or more if the authors improve the empirical evaluation and show that the current results hold with more runs.
Specifically, I want the authors to perform at least ten runs (30 would be ideal) for all experiments and show the 95% bootstrapped confidence interval.

Patterson, A., Neumann, S., White, M., & White, A. (2023). Empirical Design in Reinforcement Learning. arXiv preprint arXiv:2304.01315.

EDIT
I have updated by score based on the new results shared the authors

---

> ### Author Rebuttal · Authors · 2023-08-09
>
> Thank you for the very helpful comments. Here are our responses to the concerns raised.
>
> [W1] Evaluation could be more rigorous
>
> * Having reading the paper by Patterson et al. [1], we wholeheartedly concur that our work stands to gain from a more rigorous evaluation. To this end, during the rebuttal period we have exhausted all GPU resources that we can assess to run the full method on all HalfCheetah and Ant scenarios with 10 seeds. This process has consumed over 3 GPU-months. We assure that the remaining experiments will be performed soon and included in the revised paper. In the interim, we present the current results, accompanied by a 95% bootstrap confidence interval (around the mean), as following.
>
>   |                  | HalfCheetah performance | 95% confidence interval       | Ant performance | 95% confidence interval       |
>   | ---------------- | ----------------------- | ------------ | --------------- | ------------ |
>   | Forgetting       | 1.31 $\pm$ 0.21         | [1.11, 1.40] | 1.46 $\pm$ 0.15 | [1.36, 1.55] |
>   | Transfer         | 1.42 $\pm$ 0.19         | [1.29, 1.52] | 0.76 $\pm$ 0.07 | [0.71, 0.79] |
>   | Robustness       | 1.07 $\pm$ 0.12         | [0.98, 1.13] | 0.73 $\pm$ 0.11 | [0.68, 0.81] |
>   | Compositionality | 0.88 $\pm$ 0.09         | [0.81, 0.92] | 1.95 $\pm$ 0.11 | [1.87, 2.00] |
>   | Aggregate        | 1.17 $\pm$ 0.15         | [1.04, 1.24] | 1.22 $\pm$ 0.11 | [1.15, 1.29] |
>
>
>   And below is the updated comparison table with some baselines in [2]. It can be seen that our method remains competitive even with more training runs.
>
>   |         | HalfCheetah performance | Model size | Ant performance | Model size |
>   | ------- | ----------------------- | ---------- | --------------- | ---------- |
>   | PackNet | 0.85 $\pm$ 0.14         | 2.0        | 1.08 $\pm$ 0.21 | 2.0        |
>   | PNN     | 1.03 $\pm$ 0.14         | 8.0        | 0.98 $\pm$ 0.31 | 8.0        |
>   | FT-N    | 1.16 $\pm$ 0.20         | 8.0        | 0.97 $\pm$ 0.20 | 8.0        |
>   | CSP     | 1.27 $\pm$ 0.27         | 5.4        | 1.11 $\pm$ 0.17 | 3.9        |
>   | Ours    | 1.17 $\pm$ 0.15         | 2.1        | 1.22 $\pm$ 0.11 | 2.1        |
>
> * In order to enhance the evaluation process, we have made the following efforts: (1) In our rebuttal, the majority of comparative results are accompanied by a 95% bootstrap confidence interval (around the mean) derived from 10 individual runs. (2) The curve plots presented in our work feature both a 95% confidence interval and the median for a comprehensive analysis.
>
> [Q1] Why assume no access to previous replay buffers
>
> *  The primary concern is memory usage, as highlighted in recent literature [2,3]. Gaya et al. [2] reported that for HalfCheetah, the replay buffer used by all methods requires around 1GB per task, while for Humanoid it takes around 15GB per task. Storing experience from previous tasks would result in a large memory overhead, which is undesirable in memory-sensitive situations. Therefore, we adopt the setting without access to previous replay buffers, which also allows direct comparisons with the baselines in [2].
>
> References
>
> [1] Patterson, A., Neumann, S., White, M., & White, A. (2023). Empirical design in reinforcement learning. *arXiv preprint arXiv:2304.01315*.
>
> [2] Gaya, J. B., Doan, T., Caccia, L., Soulier, L., Denoyer, L., & Raileanu, R. (2023). Building a subspace of policies for scalable continual learning. In *International Conference on Learning Representations*.
>
> [3] Khetarpal, K., Riemer, M., Rish, I., & Precup, D. (2022). Towards continual reinforcement learning: A review and perspectives. *Journal of Artificial Intelligence Research*, *75*, 1401-1476.

---

> > ### Comment · Reviewer_CtSV · 2023-08-16
> >
> > I thank the authors for performing more runs in a short amount of time. I understand how much effort and computation it takes to do so many runs quickly. The new results in Tables 1 and 2 in the rebuttal alleviate my concerns about the statistical significance of the results. These new results with ten seeds align with those reported in the original submission. Based on these new results, I have updated my score to accept the paper. I look forward to the authors adding results with more seeds for the remaining experiments in the revised paper.

---

### Author Rebuttal · Authors · 2023-08-09

We thank all reviewers for the insightful comments, which are important for improving our work. Alongside our individual responses, we have meticulously prepared a PDF file containing figures that effectively address numerous frequently raised concerns. Below is a concise summary of these figures.

* Figure 1: Performance-size tradeoff. It includes additional comparisons with CSP (Reviewer NgVj) from 10 runs (Reviewer CtSV) and improved readability (Reviewer cvgK).

* Figure 2: Evolution of performance for multi-mode policies (Reviewers AnPE and NgVj), featuring 95% confidence interval (Reviewer CtSV) and median (Reviewer AnPE). It also uses more interpretable x labels and a revised caption (Reviewer cvgK).

---

### Decision · Program_Chairs · 2023-09-21

**Decision:**

Accept (spotlight)

**Comment:**

Meta Review for Rewiring Neurons in Non-Stationary Environments

As summarized by reviewer nWMB, this work proposes a means to efficiently expand the capacity of a neural network, namely connection permutations. The approach interleaves permutations matrices between layers of a neural network, such that input-output relationships can be adapted during learning in addition to learning the weight matrices. The authors focus in particular on RL in non-stationary environments, where adapting to new tasks can result in catastrophic forgetting of older tasks.

The permutations are learned with an existing differentiable approximation to the argsort operator. In addition, the authors propose to (a) allow the agent to cache permutations from previous tasks; (b) have the agent sample from multiple permutations to encourage exploration; and (c) align cached permutations with the latest weight matrices to ensure that previously learned policies are still consistent with the latest network, preventing catastrophic forgetting. Evaluating on a collection of continual RL tasks, the authors show SOTA or near-SOTA performance while using significantly fewer parameters than other techniques.

Most reviewers agree that this work is original, clear, and potentially significant. A clear acceptance.